# No-regret Learning with Revealed Transitions in Adversarial Markov Decision Processes

## Abstract

When learning in Adversarial Markov Decision Processes (MDPs), agents must deal with a sequence of arbitrarily chosen transition models and losses. In this paper, we consider the setting in which the transition model chosen by the adversary is revealed at the end of each episode. We propose the notion of *smoothed MDP* whose transition model aggregates with a generic function $f_t$ the ones experienced so far. Coherently, we define the concept of *smoothed regret*, and we devise Smoothed Online Mirror Descent (SOMD), an enhanced version of OMD that leverages a novel regularization term to effectively learn in this setting. For specific choices of the aggregation function $f_t$ defining the smoothed MDPs we retrieve, under full-feedback, a regret bound of order $\widetilde{\mathcal{O}}(L^{3/2}\sqrt{TL} + L\overline{C}_f^{\mathsf{P}})$ where $T$ is the number of episodes, $L$ is the horizon of the episode, and $\overline{C}_f^{\mathsf{P}}$ is a novel index of the degree of maliciousness of the adversarially chosen transitions. Under bandit feedback on the losses, we obtain a bound of order $\widetilde{\mathcal{O}}(L^{3/2}\sqrt{XAT} + L\overline{C}_f^{\mathsf{P}})$ using a simple importance weighted estimator on the losses.

## 1 Introduction

Reinforcement Learning (RL) studies sequential decision-making problems often modeled through the framework of the Markov Decision Processes (MDPs, Puterman, 2014). This framework allows the application of RL to a large variety of challenging problems and paved the way for the growing success of RL we witnessed in the last decade (e.g., Kormushev et al., 2013; Nagabandi et al., 2018; Dulac-Arnold et al., 2021). Nevertheless, MDPs are based on some grounding assumptions that may limit their modeling power in real-world scenarios. In particular, they assume that the environment dynamics $P$ (i.e., transition model) and loss[1] $\ell$ are fixed throughout the whole interaction. However, in the real world, such elements may change due to external factors which might be the effect of *nature* (e.g., system anomaly, aging effects) or *strategic* actors (e.g., adversarial attacks). While the former case is usually captured by *non-stationary* MDPs (Lecarpentier & Rachelson, 2019; Cheung et al., 2020), the latter scenario is more challenging as it assumes the presence of another agent (i.e., an adversary) acting with an objective possibly conflicting with that of the agent.

Since the early work of Even-Dar et al. (2009), this class of problems has been addressed drawing inspiration from Online Learning (OL) literature (Orabona, 2019). Adversarial Markov Decision Processes (AMDPs, Even-Dar et al., 2009) have been designed to model the scenario in which the agent interacts for $T \in \mathbb{N}$ rounds facing an adversarially chosen MDP $\mathcal{M}_t$ at every round $t \in [\![T]\!]$. Here, the performance of the agent's policy $\pi_t$ is evaluated in terms of the expected regret competing against a *fixed comparator* policy $\pi^\circ$.[2]

When the transition model $P$ is known or fixed (possibly stochastic) and the loss $\ell_t$ is adversarially chosen, several works (Zimin & Neu, 2013; Rosenberg & Mansour, 2019; Jin et al., 2020) have

---

[1]We comply with the convention of the adversarial literature of using losses instead of rewards.

[2]This is a notion of *static regret* which is different from the notion of *dynamic regret* typically adopted in non-stationary MDPs in which the comparator policy $\pi_t^\circ$ is allowed to change over rounds. It is known that, even in the simpler bandit setting, when the environment is adversarial, the no-regret property is not achievable for the dynamic regret (Bubeck et al., 2012).

achieved compelling regret guarantees of order $\widetilde{\mathcal{O}}(\sqrt{T})$. However, when the adversary is allowed to select the transition model $P_t$ at every round, the problem acquires new significant *computational* and *learning* challenges. In the *full-feedback* setting, where the transition model $P_t$ is revealed to the learner at the end of round $t$, Abbasi Yadkori et al. (2013) showed a $\widetilde{\mathcal{O}}(\sqrt{T})$ regret bound with a computationally inefficient algorithm running (a variation of) Exp3 (Auer et al., 1995) on a covering of all stochastic policies. Indeed, the computational barrier has been formalized by (Liu et al., 2022) showing that no algorithm can simultaneously achieve the no-regret property and be computationally efficient. The scenario is even less encouraging under the *bandit feedback*, where the learner observes only the collected experience. Here, Tian et al. (2021) proved a regret lower bound of order $\Omega(\min\{T, 2^H\})$, being $H$ the episode horizon. This formalizes a learning barrier showing that exponential dependence on the horizon is unavoidable.[3]

Accepting these impossibility results, the research effort has been directed towards the design of *computationally efficient* algorithms, allowing their regret to depend on the degree of adversary maliciousness in the choice of the transition model $P_t$. Inspired from the *corruption-robust* RL (Lykouris et al., 2021; Chen et al., 2021; Wei et al., 2022), the degree of maliciousness is formalized in a parameter $C^{\mathsf{P}}$ that quantifies the cumulative dissimilarity between the experienced transition models $P_t$ and a nominal one $P$:

$$C^{\mathsf{P}} := \min_{P \in \mathcal{P}} \sum_{t \in [\![T]\!]} \sum_{k \in [\![0, L-1]\!]} \max_{(x,a) \in \mathcal{X}_k \times \mathcal{A}} \|P(\cdot|x,a) - P_t(\cdot|x,a)\|_1. \tag{1}$$

In the worst case, this $C^{\mathsf{P}}$ term leads to linear regret but, in general cases, may be significantly smaller. With these premises, Jin et al. (2023) proposes a state-of-the-art algorithm working under bandit feedback for both the losses and the transition models. With the knowledge of $C^{\mathsf{P}}$, the algorithm achieves a regret dependence that gracefully degrades with the level of corruption $\widetilde{\mathcal{O}}(\sqrt{T} + C^{\mathsf{P}})$. However, when $C^{\mathsf{P}}$ is unknown, the resulting algorithm necessitates a complex inner subroutine (based on approaches from the *Corral* literature) making the final regret guarantee less explicit and, possibly, affected by large constants. Furthermore, the practicality of the algorithm in terms of computational complexity remains uncertain.

These results, therefore, leave several open questions. First, it is not clear whether the maliciousness parameter $C^{\mathsf{P}}$ inherited from the corruption-robust RL appropriately captures the challenges of the AMDP learning problem. Indeed, comparing $P_t$ against a nominal MDP does not comply with the conventional behavior of an OL algorithm, i.e., adapt according to the experience observed so far. *Can different definitions of the degree of maliciousness highlight new properties that OL algorithms for AMDPs enjoy?* Second, the approach proposed (Jin et al., 2023) aspires to directly address the bandit feedback on both the losses and transitions models. This leaves open the question of whether with transition models revealed at the end of the episode (and possibly full or bandit feedback on the losses), tighter results can be achieved, especially without the knowledge of the corruption parameter $C^{\mathsf{P}}$. *When the transition models are revealed at the end of the episode, can OL algorithms achieve better performances without the knowledge of $C^{\mathsf{P}}$?* This paper aims to address these open questions.

**Original Contributions.** The contributions of the paper can be summarized as follows.

- In Section 3, we introduce the novel concept of *smoothed* MDP and the related *smoothed* regret. Since OL algorithms make decisions based on past experience, the smoothed MDP is defined through a transition model $\overline{P}_t = f_t(P_1, \ldots, P_t)$ that aggregates with a generic function $f_t$ the previously experienced ones $P_1, \ldots, P_t$. This allows introducing a novel quantification of the adversary maliciousness $\overline{C}^{\mathsf{P}}_f$ by evaluating the dissimilarity between the chosen transition model $P_t$ and that of the smoothed MDP $\overline{P}_t$. This constant matches the corruption parameter $C^{\mathsf{P}}$ up to logarithmic terms when using an *averaging smoothing function* (i.e., $\overline{P}_t$ is the average of $P_1, \ldots, P_t$). Coherently, we define a smoothed regret measuring the learner's performance against a comparator policy $\pi^\circ$ acting in such a smoothed MDP.
- In Section 4, we propose a novel regret-minimization algorithm *Smoothed Online Mirror Descent* (SOMD). Our approach is built upon a simple yet novel instance of Online Mirror Descent (OMD) with a well-calibrated entropic regularization. Interestingly, the algorithm does not require any knowledge about $\overline{C}^{\mathsf{P}}_f$. We analyze SOMD for general smoothing functions $f_t$ and we show that

---

[3]This implies that the no-regret property is achievable only when $H = o(\log T)$ which is often unrealistic.

it achieves $\widetilde{\mathcal{O}}(L^{3/2}\sqrt{T})$ smoothed regret, which translates into $\widetilde{\mathcal{O}}(L^{3/2}\sqrt{T} + L\overline{C}_f^{\mathsf{P}})$ for the choice of the averaging smoothing function. The regret analysis requires managing non-trivial aspects, which also represents a key component of our contribution.

• In Section 5, through an importance-weighted (IW) estimator of the loss function, we extend SOMD to the bandit feedback of losses, showing comparable performances on the regret. Still, we do not require the knowledge of the degree of corruption.

## 2 PROBLEM FORMULATION

**Notation and Definitions.** In the following, given $a, b \in \mathbb{N}$ with $a \leqslant b$, we denote with $[\![a]\!] := \{1, 2, \ldots, a\}$ and with $[\![a, b]\!] := \{a, a+1, \ldots, b\}$. Given two vectors $\mathbf{x}$ and $\mathbf{y}$, we denote with $\langle \mathbf{x}, \mathbf{y} \rangle$ the inner product $\mathbf{x}^\top \mathbf{y}$. Given a set $\mathcal{A}$, we denote with $|\mathcal{A}|$ the cardinality of the set, and with $\Delta(\mathcal{A})$ the probability simplex over the set. For two generic (discrete) distributions $q, q' \in \Delta(\mathcal{A})$ we define the *negative Shannon Entropy* as $\psi(q) = \sum_{a \in \mathcal{A}} q(a) \log q(a)$ and the *Bregman Divergence* $D_\psi(q, q')$ which, for the negative Shannon Entropy, corresponds to the generalized KL-divergence $D_\psi(q, q') = \sum_{a \in \mathcal{A}} (q(a) \log(q(a)/q'(a)) - q(a) + q'(a))$. Finally, we denote with $h(p) = -p \log p - (1-p) \log(1-p)$ the binary entropy function for $p \in [0, 1]$.

**Setting.** We consider the framework of episodic loop-free Markov Decision Processes (MDPs, Puterman, 2014). Specifically, we assume the agent is interacting with a sequence of $T \in \mathbb{N}$ MDPs $\{\mathcal{M}_t\}_{t \in [\![T]\!]}$ with $\mathcal{M}_t = (\mathcal{X}, \mathcal{A}, P_t, \ell_t)$. Here, $\mathcal{X}$ is the state space, $\mathcal{A}$ is the action space, $P_t$ is the transition function $P_t : \mathcal{X} \times \mathcal{A} \to \Delta(\mathcal{X})$ such that $P_t(x'|x, a)$ is the probability of transitioning to state $x'$ after taking action $a$ in state $x$, and $\ell_t$ is the loss function $\ell_t : \mathcal{X} \times \mathcal{A} \to [0, 1]$, defined such that $\ell_t(x, a)$ is the loss the agent incurs selecting $a$ in $x$. We consider *finite* state and action sets with cardinality $|\mathcal{X}| < \infty$ and $|\mathcal{A}| = A < \infty$, respectively. As common in the literature (Jin et al., 2020; 2023) and w.l.o.g. the state space is assumed to be decomposed into $L + 1$ disjoint layers, namely $\mathcal{X} = \bigcup_{k \in [\![0, L]\!]} \mathcal{X}_k$, and $\mathcal{X}_i \cap \mathcal{X}_j = \{\}, \forall i, j \in [\![0, L]\!], i \neq j$. Furthermore, the first and the last layers are assumed to be singletons: $\mathcal{X}_0 = \{x_0\}$ and $\mathcal{X}_L = \{x_L\}$. The layered structure also imposes $P_t(x'|x, a) > 0$ only if $x \in \mathcal{X}_k$ and $x' \in \mathcal{X}_{k+1}$, for some $k \in [\![0, L-1]\!]$. Finally, to ease the exposition, we assume the cardinality of each layer to be $|\mathcal{X}_k| = X_k \leqslant X$.

**Interaction Protocol.** The interaction proceeds as follows. An adversary selects obliviously the sequence of transition models and losses $\{(P_t, \ell_t)\}_{t \in [\![T]\!]}$ before the interaction with the agent starts. Then, in each episode $t \in [\![T]\!]$, the agent sequentially decides which action to play following a *stochastic Markovian policy* $\pi_t : \mathcal{X} \to \Delta(\mathcal{A})$, where $\pi_t(a|x)$ denotes the probability of playing action $a$ in state $x$. More in detail, starting from the fixed initial state $x_{t,0} = x_0$, for each layer $k \in [\![L-1]\!]$, the agent selects the action $a_{t,k} \sim \pi_t(\cdot|x_{t,k})$, the environment evolves to the next state $x_{t,k+1} \sim P_t(\cdot|x_{t,k}, a_{t,k})$, the agent observes the loss $\ell_t(x_{t,k}, a_{t,k})$, and the interaction proceeds until the terminal state $x_{t,L} = x_L$ is reached. At the end of each episode $t$, the full transition model $P_t$ is revealed to the learner. Furthermore, in the *full-feedback model*, the full loss $\ell_t$ is revealed to the agent, while in the *bandit-feedback* model the loss is not revealed.

**Occupancy Measures.** As customary in the literature (Zimin & Neu, 2013; Jin et al., 2020), the problem will be treated in the space of *occupancy measures*. For a generic transition function $P$ and a policy $\pi$ the occupancy measure $q^{P,\pi}$ is defined as:

$$q^{P,\pi}(x, a, x') := \mathbb{P}[x_k = x, a_k = a, x_{k+1} = x'|P, \pi], \tag{2}$$

where $x \in \mathcal{X}_k$, $x' \in \mathcal{X}_{k+1}$, and $a \in \mathcal{A}$. This quantity represents the marginal probability of experiencing the transition $(x, a, x')$ when deploying policy $\pi$ in an MDP with transition model $P$. Similarly, we make use of $q^{P,\pi}(x, a) = \mathbb{P}[x_k = x, a_k = a|P, \pi] = \sum_{x' \in \mathcal{X}_{k+1}} q(x, a, x')$, and $q^{P,\pi}(x) = \mathbb{P}[x_k = x|P, \pi] = \sum_{a \in \mathcal{A}} q^{P,\pi}(x, a)$. Importantly, according to (Rosenberg & Mansour, 2019, Lemma 3.1), any valid occupancy measure is such that each layer of the MDP is visited exactly once, and thus, for every $k \in [\![0, L-1]\!]$ it holds that $\sum_{x \in \mathcal{X}_k} \sum_{a \in \mathcal{A}} \sum_{x' \in \mathcal{X}_{k+1}} q(x, a, x') = 1$, and for every $k \in [\![L-1]\!]$ and every state $x \in \mathcal{X}_k$ it holds $\sum_{x' \in \mathcal{X}_{k-1}} \sum_{a \in \mathcal{A}} q(x', a, x) = \sum_{x' \in \mathcal{X}_{k+1}} \sum_{a \in \mathcal{A}} q(x, a, x')$. This implies that the probability of entering a state coming from the previous layer is equal to the probability of leaving that same state when going to the next layer.

Finally, denoting $\Delta$ as the set of occupancies $q$ satisfying the previously defined properties, we have that every $q \in \Delta$ induces both a transition function $P^q$ and a policy $\pi^q$, computed as:

$$P^q(x'|x,a) = \frac{q(x,a,x')}{\sum_{x' \in \mathcal{X}_{k(x)+1}} q(x,a,x')}, \qquad \pi^q(a|x) = \frac{\sum_{x' \in \mathcal{X}_{k(x)+1}} q(x,a,x')}{\sum_{a \in \mathcal{A}} \sum_{x' \in \mathcal{X}_{k(x)+1}} q(x,a,x')}, \qquad (3)$$

where $k(x) \in [\![0, L]\!]$ identifies the index of the layer to which the state $x$ belongs. Throughout the analysis, when we need to refer to the occupancy at layer $k$, we use a superscript e.g., $q^k$ in favor of compactness. For a transition function $P$, we denote as $\Delta(P) \subseteq \Delta$ the set of occupancies that induce exactly the transition $P$. Similarly, given a set of transition functions $\mathcal{P}' \subseteq \mathcal{P}$, being $\mathcal{P}$ the set of all transition models, we denote as $\Delta(\mathcal{P}')$ the set of $q$'s such that $P^q \in (\mathcal{P}')$. Finally, we denote with $\Pi$ the set of Markovian stochastic policies.

**Learning Objectives.** Let $V_t^\pi(x_0) = \mathbb{E}[\sum_{k \in [\![0, L-1]\!]} \ell_t(x_{t,k}, a_{t,k})|P_t, \pi, x_0]$ be the expected cumulative loss suffered by the agent experiencing the trajectory $\{(s_{t,k}, a_{t,k})\}_{k \in [\![L-1]\!]}$ generated under the state-action distribution induced by transition function $P_t$ and policy $\pi$ in $\mathcal{M}_t$. The *expected static regret* of the agent against any comparator policy $\pi^\circ \in \Pi$ is defined as:[4]

$$\mathcal{R}_T(\pi^\circ) := \mathbb{E}\left[ \sum_{t \in [\![T]\!]} V_t^{\pi_t}(x_0) - \sum_{t \in [\![T]\!]} V_t^{\pi^\circ}(x_0) \right], \qquad (4)$$

where the expectation taken w.r.t. the internal randomization of the algorithm and on the possible stochasticity of the environment. With occupancy measures, the expected cumulative loss can be conveniently rewritten as $V_t^\pi(x_0) = \langle q^{P_t, \pi}; \ell_t \rangle$. This allows to frame the task of the agent to select occupancy measures $q_t$ instead of policies. Similarly, the expected regret can be expressed as $\mathcal{R}_T(\pi^\circ) = \mathbb{E}[\sum_{t \in [\![T]\!]} \langle q^{P_t, \pi_t} - q^{P_t, \pi^\circ}, \ell_t \rangle]$.

# 3 SMOOTHED MDPs AND SMOOTHED REGRET

As mentioned previously, our goal is to design computationally efficient algorithms for adversarial MDPs that achieve a regret scaling with a notion of the degree of maliciousness of the adversary in selecting the transition functions. However, unlike Jin et al. (2023), we aim for such a performance guarantee without introducing a notion of nominal transition function.

**Smoothed MDP.** With such an objective in mind, starting from the sequence of $T$ MDPs $\{\mathcal{M}_t\}_{t \in [\![T]\!]}$, we define a *smoothed MDP* as $\{\overline{\mathcal{M}}_t\}_{t \in [\![T]\!]}$ with $\overline{\mathcal{M}}_t = (\mathcal{X}, \mathcal{A}, \overline{P}_t, \ell_t)$ in which the transition model $P_t$ gets replaced with $\overline{P}_t$, named *smoothed transition model*. In general, $\overline{P}_t$ can be any function of the history of transition functions. Formally, for every $t \in [\![T]\!]$, let $f_t : \mathcal{P}^t \to \mathcal{P}$ and $\overline{P}_t := f_t(P_1, \ldots, P_t)$. This allows the introduction of a novel index for measuring the maliciousness of the adversary in selecting the transitions, that we call *smoothed transition error*, defined as follows:

$$\overline{C}_f^{\mathsf{P}} := \sum_{t \in [\![T]\!]} \sum_{k \in [\![0, L-1]\!]} \max_{(x,a) \in \mathcal{X}_k \times \mathcal{A}} \left\| \overline{P}_t(\cdot|x,a) - P_t(\cdot|x,a) \right\|_1. \qquad (5)$$

For adequately chosen smoothing functions, this constant interpolates between $0$ when the adversary is absent, i.e., constant transitions, and $2LT$ in the worst case. As we shall see, when $\overline{C}_f^{\mathsf{P}} = 0$, we will incur a regret of $\widetilde{\mathcal{O}}(\sqrt{T})$ even when the loss functions are completely arbitrary. To exemplify the opportunities of the smoothed transition error, we discuss the following examples.

**Example 3.1.** *Let us consider the smoothing function $f_t$ so that $\overline{P}_t = P_{t-1}$, i.e., the smoothed MDP $\overline{\mathcal{M}}_t$ has the last revealed transition model. In such a case, the smoothed transition error reduces to the* model variation $V_T$ *common in the non-stationary literature as in Cheung et al. (2020):*

$$\overline{C}_f^{\mathsf{P}} = \sum_{t \in [\![T]\!]} \sum_{k \in [\![0, L-1]\!]} \max_{(x,a) \in \mathcal{X}_k \times \mathcal{A}} \left\| P_t(\cdot|x,a) - P_{t-1}(\cdot|x,a) \right\|_1 = V_T. \qquad (6)$$

**Example 3.2** (Average Smoothing Function)**.** *Let us consider the smoothing function $f_t$ so that $\overline{P}_t = \frac{1}{t} \sum_{t' \in [\![t]\!]} P_{t'}$, i.e., the smoothed MDP $\overline{\mathcal{M}}_t$ has as transition function the average of the*

---

[4]Usually, the comparator is assumed to be the optimal policy in hindsight: $\pi^\star \in \arg\min_{\pi \in \Pi} \sum_{t \in [\![T]\!]} V_t^\pi(x_0)$.

*transition functions experienced so far. In this case, we can conveniently relate our smoothed transition error with the $C^{\mathsf{P}}$ of Jin et al. (2023):*

$$\frac{1}{\log T + 2} \leqslant \frac{\overline{C}^{\mathsf{P}}_f}{C^{\mathsf{P}}} \leqslant \log T + 2. \tag{7}$$

*This double-sided inequality is proved in Lemma A.1 and shows the equivalence of the two maliciousness measures, up to logarithmic terms.*

Together with the smoothed transition error, we introduce an index of the variability of the smoothed MDP between consecutive episodes, named *smoothed transition variability*, defined as follows:

$$\overline{D}^{\mathsf{P}}_f := \sum_{t \in [\![2,T]\!]} \max_{x,a \in \mathcal{X} \times \mathcal{A}} \left\| \overline{P}_t(\cdot|x,a) - \overline{P}_{t-1}(\cdot|x,a) \right\|_1. \tag{8}$$

**Smoothed Regret.** The notion of smoothed MDP allows us to rewrite the static regret in (4) as

$$\mathcal{R}_T(\pi^\circ) = \mathbb{E}\Big[ \underbrace{\sum_{t \in [\![T]\!]} \langle q^{\overline{P}_t, \pi_t} - q^{\overline{P}_t, \pi^\circ}; \ell_t \rangle}_{\text{Smoothed Regret } \overline{\mathcal{R}}_T(\pi^\circ)} + \underbrace{\sum_{t \in [\![T]\!]} \langle q^{P_t, \pi_t} - q^{\overline{P}_t, \pi_t}; \ell_t \rangle}_{\text{(Policy) Proxy Regret}} + \underbrace{\sum_{t \in [\![T]\!]} \langle q^{\overline{P}_t, \pi^\circ} - q^{P_t, \pi^\circ}; \ell_t \rangle}_{\text{(Comparator) Proxy Regret}} \Big].$$

The *smoothed regret* $\overline{\mathcal{R}}_T(\pi^\circ)$, accounts for the regret incurred by the agent when acting in the smoothed MDP $\overline{\mathcal{M}}$. On the other hand, the two *proxy regrets* account for the fact that both the agent's policy and the comparator policy are employed in the true MDP. Thus, the latter depends on the adversary and captures its maliciousness, while the *smoothed regret* is fully dependent on the algorithm. $\overline{\mathcal{R}}_T(\pi^\circ)$ will be treated in the next sections while the following result shows how the smoothed transition error can be employed to bound the proxy regret terms.

**Lemma 3.1** (Proxy Regret Upper Bound). *For any policy sequence $\{\pi_t\}_{t=1}^T$, and loss functions $\{\ell_t\}_{t=1}^T$ such that $\ell_t : \mathcal{X} \times \mathcal{A} \to [0,1]$ for any $t \in \{1, \ldots, T\}$ it holds that:*

$$\text{Proxy Regret} = \sum_{t \in [\![T]\!]} \langle q^{\overline{P}_t, \pi_t} - q^{P_t, \pi_t}; \ell_t \rangle \leqslant L\overline{C}^{\mathsf{P}}_f. \tag{9}$$

A reference to the proof can be found in Appendix A. Having highlighted the role of proxy regrets, in the following, we design and analyze our algorithm, namely *smoothed* OMD.

## 4 SMOOTHED OMD UNDER FULL-FEEDBACK ON LOSSES AND REVEALED TRANSITIONS

In this section, we first present a novel, smoothed, version of OMD, namely *Smoothed Online Mirror Descent* (SOMD) that exploits the intrinsic structure of smoothed MDPs, and, then, we provide the analysis of its no-smooth-regret property. We start considering generic smoothing functions $f_t$, then we focus on the average smoothing function (Example 3.2).

**Algorithm Design.** Coherently with the OMD algorithmic blueprint, at the end of each episode $t \in [\![T]\!]$, SOMD (Algorithm 1) computes an occupancy measure $q_{t+1}$ that trades off between minimizing the loss of the round $\ell_t$ and not diverging excessively from a specific regularization reference $\overline{q}_t$. In mathematical terms, SOMD solves the constrained convex program:

$$q_{t+1} = \underset{q \in \Delta(\overline{P}_t)}{\arg\min} \langle q, \ell_t \rangle + \frac{1}{\eta} D_\psi(q, \overline{q}_t), \tag{10}$$

where $\eta > 0$ is a regularization hyperparameter. It is worth noting that the resulting occupancy $q_{t+1}$ is constrained into $\Delta(\overline{P}_t)$, i.e., the set of occupancies realizable with the smoothed transition model $\overline{P}_t$. More importantly, the resulting policy $\pi_{t+1} := \pi^{q_{t+1}}$ will be evaluated in the environment paired with the smoothed transition model $\overline{P}_{t+1}$, leading to the occupancy $q^{\overline{P}_{t+1}, \pi_{t+1}}$. In general, we have that $q^{\overline{P}_{t+1}, \pi_{t+1}} \neq q_{t+1} = q^{\overline{P}_t, \pi_{t+1}}$. Intuitively, our program in Equation (10) "pretends" that $\pi_{t+1}$ will be played in the current smoothed MDP $\overline{P}_t$, instead it will be played in $\overline{P}_{t+1}$. This

---

**Algorithm 1:** *Smoothed* **O**nline **M**irror **D**escent in Full-Feedback

---

**Input :** state space $\mathcal{X}$, action space $\mathcal{A}$, episode number $T$, learning rate $\eta > 0$, mixing parameter $\alpha \in [0, 1]$, smoothing functions $\{f_t\}_{t \in [\![T]\!]}$.

**Initialize :** Set $\pi_1 = \pi^{q_1}$, $q_1(x, a, x') = u(x, a, x')$ $\forall k \in [\![L-1]\!], (x, a, x') \in \mathcal{X}_k \times \mathcal{A} \times \mathcal{X}_{k+1}$.

1  **for** $t = 1, \ldots, T$ **do**
2    Execute policy $\pi_t$ in $\mathcal{M}_t$ and observe $(P_t, \ell_t)$.
3    Compute *smoothed* transition $\overline{P}_t = f_t(P_1, \ldots, P_t)$.
4    Compute *smoothed* regularization point $\overline{q}_t = (1 - \alpha) q_t + \alpha u$
5    Perform mirror descent step $q_{t+1} = \arg\min_{q \in \Delta(\overline{P}_t)} \langle q, \ell_t \rangle + \frac{1}{\eta} D_\psi(q, \overline{q}_t)$
6    Update policy $\pi_{t+1} = \pi^{q_{t+1}}$
7  **end**

---

represents a fundamental feature of SOMD that deviates from the classical OMD algorithms, which are often designed to deal with a fixed decision set $\Delta(P)$ (Jin et al., 2020) or a sequence of nested sets (Jin et al., 2023). SOMD manages the mismatch between such domains by: $(i)$ leveraging the transition variability of smooth MDPs, encoded in term $\overline{D}_f^{\mathsf{P}}$; $(ii)$ computing a *smoothed regularization reference*, $\overline{q}_t$, defined as a mixture between the SOMD decision at the previous step, $q_t$, and the uniform occupancy measure $u(x, a, x') = 1/(X_k A X_{k+1})$ for every $k \in [\![0, L-1]\!]$. Specifically, $\overline{q}_t = (1 - \alpha) q_t + \alpha u$, where $\alpha \in [0, 1]$ is a hyperparameter to be specified later that acts as a further source of regularization.[5] The choice of $\alpha > 0$ will generate a bias term whose effect on the regret will be controlled through a proper selection of the value of $\alpha$. Clearly, as supported by intuition, the solution delivered by the program in Equation (10) will be a good representative of the actual occupancy only when two consecutive smoothed MDPs are sufficiently similar $\overline{P}_t \approx \overline{P}_{t+1}$. This is encoded in the variability term $\overline{D}_f^{\mathsf{P}}$ that emerges in the regret analysis.

**Smoothed Regret Analysis.** The following provides the smoothed regret upper bound for SOMD.

**Theorem 4.1** (Smoothed-Regret Bound for SOMD under full-feedback). *Let* $\eta = \sqrt{(10L \log(2X^2 AT) \rho_T^f)/T}$ *and* $\alpha = 1/(1 + T)$ *, then for any comparator policy* $\pi^\circ \in \Pi$ *Algorithm 1 suffers a smoothed regret of:*

$$\overline{\mathcal{R}}_T(\pi^\circ) \leqslant \mathcal{O}\left( L^2 \overline{D}_f^{\mathsf{P}} + L^{3/2} \sqrt{T \log(X^2 AT) \rho_T^f} \right), \tag{11}$$

*where* $\rho_T^f := \log(T) + \overline{D}_f^{\mathsf{P}} + \mathcal{H}_T(\overline{D}_f^{\mathsf{P}})$ *and* $\mathcal{H}_T(\overline{D}_f^{\mathsf{P}}) = TLh\left( \frac{(L^2 + L)\overline{D}_f^{\mathsf{P}}}{2TL} \right)$.

As one could expect, the behavior of the constant $\mathcal{H}_T(\overline{D}_f^{\mathsf{P}})$ depends on the choice of the smoothing function to be used. For specific choices of smoothing functions, as it is for average smoothing functions, this term will be sub-linear in $T$. For the average choice of Example 3.2, we have that $\mathcal{H}_T(\overline{D}_f^{\mathsf{P}}) = O(L^2 \log(T)^2)$; the interested reader can refer to Appendix C for in-detail analysis.

Now, we provide the reader with additional insight into our approach, by specializing the analysis and results to the specific class of smoothing functions that satisfy $\overline{P}_t := \frac{1}{t} \sum_{t' \in [\![t]\!]} P_{t'}$ called "average smoothing function". With such an additional structure, our algorithm is able to guarantee,

**Corollary 4.2** (Smoothed-Regret Bound for SOMD under full-feedback and average smoothing). *Let* $\eta = 3\sqrt{(2L \log(2X^2 AT) \log(T))/T}$, $\alpha = 1/(T + 1)$ *and smoothing functions such that* $\overline{P}_t := \frac{1}{t} \sum_{t' \in [\![t]\!]} P_{t'}$, *for any comparator policy* $\pi^\circ \in \Pi$, *Algorithm 1 suffers a smoothed regret of:*

$$\overline{\mathcal{R}}_T(\pi^\circ) \leqslant \mathcal{O}\left( L^2 \log(T) + L^{3/2} \sqrt{T \log(X^2 AT) \log(T)} \right). \tag{12}$$

While the full derivation can be found in Appendix B, here we outline the most relevant steps.

*Proof Sketch.* From now on we will overload the notation with $q_t^{\overline{P}_{t-1}, \pi_t} = q_t$ and $q_t^{\overline{P}_t, \pi^\circ} = q_t^\circ$ for compactness. SOMD computes $q_t$ based on $\overline{P}_{t-1}$. Thus we isolate the part of the regret solely

---

[5] Parameter $\alpha$ is needed for technical reasons that will become clear in the proof of Theorem 4.1.

dependent on this algorithmic choice:

$$\overline{\mathcal{R}}_T(\pi^\circ) = \mathbb{E}\Big[\sum_{t=1}^T \langle q^{\overline{P}_t,\pi_t} - q^{\overline{P}_t,\pi^\circ}; \ell_t\rangle\Big] = \underbrace{\mathbb{E}\Big[\sum_{t=1}^T \langle q_t - q_t^\circ; \ell_t\rangle\Big]}_{\text{Algorithmic Regret } (\overline{\mathcal{R}}_T^A)} + \underbrace{\mathbb{E}\Big[\sum_{t=1}^T \langle q^{\overline{P}_t,\pi_t} - q^{\overline{P}_{t-1},\pi_t}; \ell_t\rangle\Big]}_{\text{Update Regret } (\overline{\mathcal{R}}_T^U)}.$$

The *Update Regret* captures the mismatch in using $\overline{P}_{t-1}$ to compute $\pi^{q_t}$ and then using the same policy in $\overline{P}_t$. This term is affected by the magnitude of $\overline{D}_f^{\mathsf{P}}$, thus by the "slowly"-changing behavior of the smoothed MDP, as analyzed in Lemma B.1. The *Algorithmic Regret* is the one specifically controlled by the algorithm, and it can be decomposed into the following terms:

$$\overline{\mathcal{R}}_T^A = \underbrace{\mathbb{E}\Big[\sum_{t=1}^T \langle \overline{q}_t - q_t^\circ; \ell_t\rangle\Big]}_{\text{Descent Regret } (\overline{\mathcal{R}}_T^D)} + \underbrace{\mathbb{E}\Big[\sum_{t=1}^T \langle q_t - \overline{q}_t; \ell_t\rangle\Big]}_{\text{Regularization Regret } (\overline{\mathcal{R}}_T^R)}.$$

where the *Regularization Regret* describes the degree to which $\overline{q}_t$ is different from $q_t$ and it is controlled by the mixing coefficient $\alpha$ (Lemma B.3). The *Descent Regret* captures how performing an OMD step in the smoothed MDP affects the overall performance and satisfies:

$$\eta\overline{\mathcal{R}}_T^D \leqslant \mathbb{E}\Big[\underbrace{\sum_{t=1}^T D_\psi(\overline{q}_t, \tilde{q}_{t+1})}_{\text{"Stability" term}} + \underbrace{\frac{\alpha}{(1-\alpha)}\sum_{t=1}^T D_\psi(q_t^\circ, u)}_{\text{"Residual" term}} + \underbrace{\sum_{t=1}^T (D_\psi(q_t^\circ, \overline{q}_t) - D_\psi(q_t^\circ, \overline{q}_{t+1}))}_{\text{"Penalty" term}}\Big],$$

where $\tilde{q}_{t+1}$ is the solution to the unconstrained OMD problem, as can be seen in Lemma B.4. The related "Stability" term can be bounded by standard OMD analysis as done in Jin et al. (2020). The "Residual" term catches the effect of mixing the output of the descent step with uniform distributions but it can be bounded pretty easily as in Lemma B.9. Finally and more interestingly, the "Penalty" term is due to the presence of a regularizer in optimizing the cumulative loss (see Lemma B.10). In standard OL analysis, a similar term can be easily bounded using telescoping arguments. However, the time-varying nature of smooth MDPs prevents us from using such arguments and bounding this term requires some machinery. Specifically, we first rewrite the penalty term as,

$$\sum_{t=1}^T (D_\psi(q_t^\circ, \overline{q}_t) - D_\psi(q_t^\circ, \overline{q}_{t+1})) \leqslant D_\psi(q_1^\circ, \overline{q}_1) + \sum_{t=2}^T D_\psi(q_t^\circ, \overline{q}_t) - D_\psi(q_{t-1}^\circ, \overline{q}_t) \qquad (13)$$

While the first term can be easily bounded by the maximum range of the regulariser, as shown in Lemma B.11, the second term requires more machinery despite its non-telescoping behavior. First, we leverage the properties of the KL-divergence to obtain:

$$\sum_{t=2}^T D_\psi(q_t^\circ, \overline{q}_t) - D_\psi(q_{t-1}^\circ, \overline{q}_t) \leqslant \sum_{t=2}^T |\psi(q_t^\circ) - \psi(q_{t-1}^\circ)| + \sum_{t=2}^T \|\log(\overline{q}_t)\|_\infty \|q_t^\circ - q_{t-1}^\circ\|_1.$$

The second summation can be easily bounded via Lemmas B.21 and B.20. For the first summation, we employ Theorem 3 by Sason (2013) to bound the absolute entropy differences: we first identify two different time regimes separated by $\bar{t} := \lceil L/1 - \frac{1}{X^2 A} \rceil$. Now, for $t < \bar{t}$ we simply apply the above-cited theorem. For $t \geqslant \bar{t}$ instead, the use of averaging as smoothing function allows us to further bound the total variation distance $d_{TV}(q_t^{\circ,k}, q_{t-1}^{\circ,k}) \leqslant L/t$ as for Lemma F.4. This allows us to bound the summation with sub-linear terms:

$$\sum_{t=\bar{t}}^T |\psi(q_t^\circ) - \psi(q_{t-1}^\circ)| \leqslant \sum_{t=\bar{t}}^T \sum_{k=0}^{L-1} h(\epsilon_t) + \log(X^2 AT + X^2 A)\epsilon_t$$

Finally, optimizing for $\eta$ in the *Algorithmic Regret* and combining all the single terms in the decomposition returns the final result. $\qquad\square$

**Overall Regret Analysis.** Now that we have proven that Algorithm 1 is no-smooth regret, what is simply left do is to combine this result with Lemma 3.1, leading to the following result.

---

**Algorithm 2:** *Smoothed Online Mirror Descent in Bandit-Feedback*

---

**Input :** state space $\mathcal{X}$, action space $\mathcal{A}$, episode number $T$, learning rate $\eta > 0$, mixing parameter $\alpha \in [0, 1]$, estimator parameter $\gamma > 0$, smoothing functions $\{f_t\}_{t \in \llbracket T \rrbracket}$.

**Initialize :** Set $\pi_1 = \pi^{q_1}$, $q_1(x, a, x') = u(x, a, x')$ $\forall k \in \llbracket 0, L-1 \rrbracket$, $(x, a, x') \in \mathcal{X}_k \times \mathcal{A} \times \mathcal{X}_{k+1}$.

**1 for** $t = 1, \dots, T$ **do**

  **2**    Execute policy $\pi_t$ in $\mathcal{M}_t$ and collect trajectory $\{(x_{t,k}, a_{t,k}, \ell_t(x_{t,k}, a_{t,k}))\}_{t \in \llbracket T \rrbracket, k \in \llbracket 0, L-1 \rrbracket}$

  **3**    Observe $P_t$ and compute *smoothed* transition $\overline{P}_t = f_t(P_1, \dots, P_t)$.

  **4**    Construct loss estimator: $\hat{\ell}_t(x, a) = \frac{\ell_t(x,a)}{q^{P_t, \pi_t}(x,a) + \gamma} \mathbb{1}_t(x, a), \quad \forall (x, a) \in \mathcal{X} \times \mathcal{A}$

  **5**    Compute *smoothed* regularization point $\overline{q}_t = (1 - \alpha) q_t + \alpha u$

  **6**    Perform mirror descent step $q_{t+1} = \arg\min_{q \in \Delta(\overline{P}_t)} \langle q, \hat{\ell}_t \rangle + \frac{1}{\eta} D_\psi(q, \overline{q}_t)$

  **7**    Update policy $\pi_{t+1} = \pi^{q_{t+1}}$

**8 end**

---

**Corollary 4.3** (Regret Bound for SOMD under full-feedback and average smoothing)**.** *For* $\eta = 3\sqrt{(2L \log(2X^2AT) \log(T))/T}$, $\alpha = 1/(T+1)$, *smoothing functions such that* $\overline{P}_t := \frac{1}{t} \sum_{t' \in \llbracket t \rrbracket} P_{t'}$ *and any comparator policy* $\pi^\circ \in \Pi$, *Algorithm 1 suffers a regret of:*

$$\mathcal{R}_T(\pi^\circ) \leqslant \widetilde{\mathcal{O}} \left( L^{3/2}\sqrt{T} + L\overline{C}_f^{\mathsf{P}} \right). \tag{14}$$

## 5   SMOOTHED OMD UNDER BANDIT-FEEDBACK ON LOSSES AND REVEALED TRANSITIONS

In this section, we extend the previous results to the case in which losses are observed under bandit feedback. We show that SOMD can be adapted to this setting with limited adjustments.

**Algorithmic Design.** To face this challenging scenario, a common way to go is to construct loss estimators based on observations only. In particular, inverse importance-weighted estimators as of Jaksch et al. (2010) can be used to weight the estimation on the experienced trajectory. Thus, we will simply substitute the true feedback with an estimator, namely:

$$\hat{\ell}_t(x, a) := \frac{\ell_t(x, a)}{q^{P_t, \pi_t}(x, a) + \gamma} \mathbb{1}_t(x, a), \tag{15}$$

where $\gamma > 0$ is a parameter to be specified later that allows bounding the variance of the estimator, and $\mathbb{1}_t(x, a)$ is the indicator random variable for the event that the $(x, a)$ is visited at round $t$. As it emerges from the analysis, the SOMD algorithm can be employed by just replacing $\ell_t$ with $\hat{\ell}_t$. The intrinsic properties of smoothing in smoothed MDPs will take care of most of the remaining complexity of the problem and the rest of the SOMD algorithm can be employed as is (Algorithm 2).

**Smoothed Regret Analysis.** We again proceed in bounding the regret in the smoothed MDP. In particular, we can state that using generic smoothing functions leads to the following:

**Theorem 5.1** (Smoothed-Regret Bound for SOMD under bandit-feedback)**.** *Let* $\eta = \sqrt{(13L \log(2X^2AT)\rho_T^f)/(2XAT)}$, $\alpha = 1/(T+1)$, $\gamma = \eta$, *generic smoothing functions and any comparator policy* $\pi^\circ \in \Pi$, *Algorithm 2 suffers a smoothed regret of:*

$$\overline{\mathcal{R}}_T(\pi^\circ) \leqslant \mathcal{O} \left( L^2 \overline{D}_f^{\mathsf{P}} + L\overline{C}_f^{\mathsf{P}} + L^{3/2}\sqrt{XAT \rho_T^f \log(X^2AT)} \right). \tag{16}$$

The corresponding result with the choice of the average smoothing is the following.

**Corollary 5.2** (Smoothed-Regret Bound for SOMD under bandit-feedback and average smoothing)**.** *Let* $\eta = \sqrt{(21L \log(2X^2AT)(\log(T)))/(2XAT)}$, $\alpha = 1/(T+1)$, $\gamma = \eta$, *smoothing functions such that* $\overline{P}_t := \frac{1}{t} \sum_{t' \in \llbracket t \rrbracket} P_{t'}$ *and any comparator policy* $\pi^\circ \in \Pi$, *Algorithm 2 suffers a smoothed regret of:*

$$\overline{\mathcal{R}}_T(\pi^\circ) \leqslant \mathcal{O} \left( L\overline{C}_f^{\mathsf{P}} + L^2 \log(T) + L^{3/2}\sqrt{XAT \log(X^2AT) \log(T)} \right) \tag{17}$$

The proof for such a result builds upon the same structure of the full feedback case and we refer the reader to Appendix D for a detailed explanation. Interestingly, and differently from the full-feedback setting, the smoothing transition error $\overline{C}_f^{\mathsf{P}}$ term appears even in the *smoothed regret*. This is inherited by the bias of the loss estimator $\hat{\ell}_t$ of Equation 15.

**Overall Regret Analysis.** As for the full-feedback case, now that we have proven the smoothed regret bound for Algorithm 2 we combine this result with Lemma 3.1, leading to the following:

**Corollary 5.3** (Regret Bound for SOMD under bandit-feedback and average smoothing)**.** *Let $\eta = \sqrt{(21L\log(2X^2AT)(\log(T)))/(2XAT)}$, $\alpha = 1/(T+1)$, $\gamma = \eta$, smoothing functions such that $\overline{P}_t := \frac{1}{t}\sum_{t'\in[\![t]\!]} P_{t'}$ and any comparator policy $\pi^\circ$, Algorithm 2 suffers a regret of:*

$$\mathcal{R}_T(\pi^\circ) = \tilde{\mathcal{O}}\left(L\overline{C}_f^{\mathsf{P}} + L^{3/2}\sqrt{XAT}\right). \tag{18}$$

## 6 RELATED WORKS

The presence of disturbances, adversarial attacks, and non-stationary behaviors in MDPs have been extensively treated through various lenses. We now highlight how this work is related to each subfield.

**Adversarial MDPs.** When only the losses are adversarially chosen and the transition functions are either known or fixed (i.e., $\overline{C}_f^{\mathsf{P}} = 0$), many are cases of success in obtaining compelling regret guarantees (e.g. Zimin & Neu, 2013; Rosenberg & Mansour, 2019; Jaksch et al., 2010; Jin et al., 2020), where Zimin & Neu (2013); Jin et al. (2020) in particular take advantage of performing OMD steps in the occupancy measure space. Under adversarially chosen transitions as well, the only related work up to our knowledge is Jin et al. (2023), where bandit feedback for both the losses and the transition model is considered. However, when the degree of maliciousness is unknown, the resulting computational complexity remains uncertain and the regret guarantees are rather implicit. Furthermore, Jin et al. (2023) does not take into account the intermediate setting of revealed transitions, but applying their Algorithm 1 to such a setting would lead to a regret of $\mathcal{O}(L^2 X\sqrt{AT\log(LXAT^2)} + L^5 X^4 A\log(LXAT^2) + C^{\mathsf{P}} L^5 X^4 \log(LXAT^2))$ *even with a known degree of maliciousness $C^{\mathsf{P}}$,* which is significantly worse than the performances of SOMD, constant-wise. On the other hand, we positively leverage the intermediate setting of revealed transitions, introducing ad-hoc degrees of maliciousness the notion of smoothed MDP formalism, and finally recovering comparable performances via computationally efficient OMD-based algorithms.

**Corruption Robust RL.** Works in this line (e.g. Lykouris et al., 2021; Chen et al., 2021) typically assume the presence of an adversary corrupting some of the rewards and/or transitions, compared to a nominal underlying MDP. These works then address a different notion of regret, namely the one defined coherently with respect to the loss incurred by the best policy in the nominal MDP and denoting the number of corrupted episodes by $C$, Wei et al. (2022) is the first to achieve a regret of $\tilde{\mathcal{O}}(\min\{\frac{1}{\Delta}, \sqrt{T}\} + C)$ in a bandit feedback setting without requiring the knowledge of $C$, with $\Delta$ being the reward gap between the best and the second-best.

**Robust MDPs & RL.** Robust MDPs (e.g. Nilim & Ghaoui, 2005; Wiesemann et al., 2013) and Robust Reinforcement Learning (e.g. Morimoto & Doya, 2005; Lim et al., 2016) focus on computing policies that exhibit robustness in face of uncertainties over the transition and/or loss models so that to withstand potential mismatches between the models and the ground truth. Usually, though, minimax solutions against the worst-case scenario are sought, failing to adapt to easier instances and to smoothly interpolate performance based on the degree of mismatch.

**Non-Stationary RL.** Works in this line (e.g. Lecarpentier & Rachelson, 2019; Wei & Luo, 2021; Cheung et al., 2023) allow the MDP model to change arbitrarily over time and the performance metric employed is the dynamic regret, i.e. competing against a comparator policy varying over rounds. It is known that, even in the simpler bandit setting, when the environment is adversarial the no-regret property is not achievable for the dynamic regret (Bubeck et al., 2012), so the results presented in this work are not directly comparable with them.

## 7 DISCUSSION AND CONCLUSIONS

In this paper, we have addressed the OL problem in MDPs in which the transition functions are chosen by an adversary. Starting from the known computational and statistical limits of this challenging setting, we have the scenario in which the transition functions are revealed at the end of the episode. We have introduced the notion of smoothed MDP and, based on it, we designed suitable indexes $\overline{C}_f^{\mathsf{P}}$ and $\overline{D}_f^{\mathsf{P}}$ to assess the degree of maliciousness of the adversary that relate and generalize the existing ones. These indexes allowed us to design a computationally efficient algorithm, SOMD, that enjoys regret guarantees of order $\tilde{\mathcal{O}}(\sqrt{T} + \overline{C}_f^{\mathsf{P}})$. These results are in line with the literature (Jin et al., 2023) but require no knowledge of the maliciousness index $\overline{C}_f^{\mathsf{P}}$ and are obtained with simple yet computationally efficient algorithms. Future works include the extension of the proposed approach to a *complete bandit feedback* setting where the transition functions are not revealed at the end of the episode. Furthermore, their generality makes our indexes $\overline{C}_f^{\mathsf{P}}$ and $\overline{D}_f^{\mathsf{P}}$ suitable to be employed beyond the specific averaging smooth of Example 3.2 and capture more sophisticated relations in the sequence of the transition models, such as *bounded variation* (Example 3.1).

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

## A    THEORETICAL ANALYSIS FOR SMOOTHED MDPS

**Lemma A.1.** *Let $\overline{P}_t = \frac{1}{t} \sum_{t' \in [\![t]\!]} P_{t'}$. Then, it holds that:*

$$\frac{1}{\log T + 2} \leqslant \frac{\overline{C}_f^{\mathsf{P}}}{C^{\mathsf{P}}} \leqslant \log T + 2. \tag{19}$$

*Proof.* Let $P$ be the nominal MDP with which $C^{\mathsf{P}}$ is defined. We have:

$$\begin{aligned}
\overline{C}_f^{\mathsf{P}} &= \sum_{t \in [\![T]\!]} \sum_{k \in [\![0,L-1]\!]} \max_{(x,a) \in \mathcal{X}_k \times \mathcal{A}} \left\| \overline{P}_t(\cdot|x,a) - P_t(\cdot|x,a) \right\|_1 \\
&\leqslant \sum_{t \in [\![T]\!]} \sum_{k \in [\![0,L-1]\!]} \max_{(x,a) \in \mathcal{X}_k \times \mathcal{A}} \left\| \overline{P}_t(\cdot|x,a) - P(\cdot|x,a) \right\|_1 \\
&\quad + \sum_{t \in [\![T]\!]} \sum_{k \in [\![0,L-1]\!]} \max_{(x,a) \in \mathcal{X}_k \times \mathcal{A}} \left\| P(\cdot|x,a) - P_t(\cdot|x,a) \right\|_1 \\
&= \sum_{t \in [\![T]\!]} \sum_{k \in [\![0,L-1]\!]} \max_{(x,a) \in \mathcal{X}_k \times \mathcal{A}} \left\| \frac{1}{t} \sum_{t' \in [\![t]\!]} P_{t'}(\cdot|x,a) - P(\cdot|x,a) \right\|_1 + C^{\mathsf{P}} \\
&\leqslant \sum_{t \in [\![T]\!]} \frac{1}{t} \sum_{t' \in [\![t]\!]} \sum_{k \in [\![0,L-1]\!]} \max_{(x,a) \in \mathcal{X}_k \times \mathcal{A}} \left\| P_{t'}(\cdot|x,a) - P(\cdot|x,a) \right\|_1 + C^{\mathsf{P}} \\
&\leqslant \sum_{t \in [\![T]\!]} \frac{1}{t} \sum_{t' \in [\![T]\!]} \sum_{k \in [\![0,L-1]\!]} \max_{(x,a) \in \mathcal{X}_k \times \mathcal{A}} \left\| P_{t'}(\cdot|x,a) - P(\cdot|x,a) \right\|_1 + C^{\mathsf{P}} \\
&\leqslant (\log T + 2) C^{\mathsf{P}},
\end{aligned}$$

where the first inequality comes from triangle inequality. And the second from Jensen's. The lower bound comes from analogous derivation. □

**Lemma 3.1** (Proxy Regret Upper Bound). *For any policy sequence $\{\pi_t\}_{t=1}^T$, and loss functions $\{\ell_t\}_{t=1}^T$ such that $\ell_t : \mathcal{X} \times \mathcal{A} \to [0,1]$ for any $t \in \{1, \ldots, T\}$ it holds that:*

$$\textit{Proxy Regret} = \sum_{t \in [\![T]\!]} \langle q^{\overline{P}_t, \pi_t} - q^{P_t, \pi_t}; \ell_t \rangle \leqslant L \overline{C}_f^{\mathsf{P}}. \tag{9}$$

*Proof.* The proof follows the same steps as in the original derivation of Lemma F.7 in (Jin et al., 2023), we invite the reader to see the original work for a detailed proof. □

## B    THEORETICAL ANALYSIS WITH FULL-FEEDBACK AND AVERAGE SMOOTHING FUNCTIONS

In this section, we present the proofs of the results discussed in Section 4 with the specific smoothing functions $f_t(P_1, \ldots, P_t) = \frac{1}{t} \sum_{t'=1}^t P_{t'}$.

### B.1    MAIN RESULTS

**Corollary 4.2** (Smoothed-Regret Bound for SOMD under full-feedback and average smoothing). *Let $\eta = 3\sqrt{(2L \log(2X^2 AT) \log(T))/T}$, $\alpha = 1/(T+1)$ and smoothing functions such that $\overline{P}_t := \frac{1}{t} \sum_{t' \in [\![t]\!]} P_{t'}$, for any comparator policy $\pi^\circ \in \Pi$, Algorithm 1 suffers a smoothed regret of:*

$$\overline{\mathcal{R}}_T(\pi^\circ) \leqslant \mathcal{O}\left( L^2 \log(T) + L^{3/2} \sqrt{T \log\left(X^2 AT\right) \log\left(T\right)} \right). \tag{12}$$

*Proof.* We first define $q_t^{\overline{P}_{t-1}, \pi_t} = q_t$ and $q_t^{\overline{P}_t, \pi^\circ} = q_t^\circ$ for notational simplicity. It follows that the smoothed regret term can be decomposed as

$$\overline{\mathcal{R}}_T(\pi^\circ) = \mathbb{E}\Big[\sum_{t=1}^T \langle q^{\overline{P}_t, \pi_t} - q^{\overline{P}_t, \pi^\circ}; \ell_t \rangle\Big] = \underbrace{\mathbb{E}\Big[\sum_{t=1}^T \langle q_t - q_t^\circ; \ell_t \rangle\Big]}_{\text{Algorithmic Regret } (\overline{\mathcal{R}}_T^A)\atop \text{Lemma B.2}} + \underbrace{\mathbb{E}\Big[\sum_{t=1}^T \langle q^{\overline{P}_t, \pi_t} - q^{\overline{P}_{t-1}, \pi_t}; \ell_t \rangle\Big]}_{\text{Update Regret } (\overline{\mathcal{R}}_T^U)\atop \text{Lemma B.1}},$$

Then, the result follows directly from the combination of Lemma B.1 for the Update Regret and Lemma B.2 for the Algorithmic Regret, namely:

$$\overline{\mathcal{R}}_T(\pi^\circ) \leqslant 2(L^2 + L)(1 + \log(T)) + 2L + 8L\sqrt{TL \log(2X^2 AT) \log(T)}$$
$$= 4L + 2L^2 + 2(L^2 + L)(1 + \log(T)) + 6L\sqrt{2TL \log(2X^2 AT) \log(T)}$$

which leads to the final result. $\qquad\square$

**Corollary 4.3** (Regret Bound for SOMD under full-feedback and average smoothing). *For $\eta = 3\sqrt{(2L \log(2X^2 AT) \log(T))/T}$, $\alpha = 1/(T+1)$, smoothing functions such that $\overline{P}_t := \frac{1}{t} \sum_{t' \in [\![t]\!]} P_{t'}$ and any comparator policy $\pi^\circ \in \Pi$, Algorithm 1 suffers a regret of:*

$$\mathcal{R}_T(\pi^\circ) \leqslant \tilde{\mathcal{O}}\left(L^{3/2}\sqrt{T} + L\overline{C}_f^{\mathsf{P}}\right). \tag{14}$$

*Proof.* The result follows straightforwardly from combining the results from Corollary 4.2 and Lemma 3.1. $\qquad\square$

**Lemma B.1** (Update Regret Bound). *For smoothing functions such that $\overline{P}_t := \frac{1}{t} \sum_{t' \in [\![t]\!]} P_{t'}$ and $\ell_t : \mathcal{X} \times \mathcal{A} \to [0, 1]$, for any policy sequence $\{\pi_t\}_{t \in [\![T]\!]}$ we have that,*

$$\overline{\mathcal{R}}_T^U = \mathbb{E}\Big[\sum_{t=1}^T \langle q^{\overline{P}_t, \pi_t} - q^{\overline{P}_{t-1}, \pi_t}; \ell_t \rangle\Big] \leqslant 2(L^2 + L)(1 + \log(T)).$$

*Proof.*

$$\overline{\mathcal{R}}_T^U = \mathbb{E}\Big[\sum_{t=1}^T \langle q^{\overline{P}_t, \pi_t} - q^{\overline{P}_{t-1}, \pi_t}; \ell_t \rangle\Big] \leqslant \mathbb{E}\Big[\sum_{t=1}^T \left\|q^{\overline{P}_t, \pi_t} - q^{\overline{P}_{t-1}, \pi_t}\right\|_1\Big]$$

$$\leqslant 2(L^2 + L)\sum_{t=1}^T \frac{1}{t}$$

$$\leqslant 2(L^2 + L)(\log(T) + 1)$$

The first inequality follows from Holder's and the fact that $\|\ell_t\|_\infty \leqslant 1$ by definition. The second inequality follows from Lemma F.10, and the last one comes from the bound on the harmonic series. $\qquad\square$

**Lemma B.2** (Algorithmic Regret Bound). *Choosing $\eta = 3\sqrt{\frac{2L \log(2X^2 AT) \log(T)}{T}}$, $\alpha = \frac{1}{1+T}$ Algorithmic Regret is bounded by*

$$\overline{\mathcal{R}}_T^A = \mathbb{E}\Big[\sum_{t=1}^T \langle q_t - q_t^\circ; \ell_t \rangle\Big] \leqslant 2L + 6L\sqrt{TL \log(2X^2 AT) \log(T)}$$

*Proof.* The proof relies on the following decomposition:

$$\eta\overline{\mathcal{R}}_T^A = \eta\mathbb{E}\Big[\sum_{t=1}^T \langle q_t - q_t^\circ; \ell_t \rangle \pm \eta\sum_{t=1}^T \langle \overline{q}_t; \ell_t \rangle\Big] = \eta\underbrace{\mathbb{E}\Big[\sum_{t=1}^T \langle \overline{q}_t - q_t^\circ; \ell_t \rangle\Big]}_{\text{Descent Regret } (\overline{\mathcal{R}}_T^D)\atop \text{Lemma B.4}} + \eta\underbrace{\mathbb{E}\Big[\sum_{t=1}^T \langle q_t - \overline{q}_t; \ell_t \rangle\Big]}_{\text{Regularization Regret } (\overline{\mathcal{R}}_T^R)\atop \text{Lemma B.3}}.$$

By applying Lemma B.3 for the Regularization Regret and Lemma B.4 for the Descent Regret we can rewrite the Algorithmic Regret as

$$\eta \overline{\mathcal{R}}_T^A \leqslant 2\eta L + \eta^2 T L + 18 L^2 \log\left(2X^2 A T\right) \log\left(T\right)$$

We first chose the optimal $\eta$ as:

$$\eta = 3\sqrt{\frac{2L \log(2X^2 A T) \log(T)}{T}}$$

and then we upper bound the Algorithmic Regret after substituting the optimal $\eta$, namely:

$$\overline{\mathcal{R}}_T^A \leqslant 2L + \eta T L + \frac{18 L^2 \log\left(2X^2 A T\right) \log(T)}{\eta}$$
$$\leqslant 2L + 3L\sqrt{2TL \log\left(2X^2 A T\right) \log\left(T\right)} + 3L\sqrt{2TL \log\left(2X^2 A T\right) \log(T)}$$

which leads to the final result. $\qquad\square$

**Lemma B.3** (Regularisation Regret Bound). *For $\overline{q}_t = (1-\alpha) q_t + \alpha u, \quad u(x, a, x') = \frac{1}{X_k \cdot A \cdot X_{k+1}} \ \forall k \in \llbracket L-1 \rrbracket$ and $\alpha = \frac{1}{T+1}$, it holds that*

$$\eta \overline{\mathcal{R}}_T^R = \eta \mathbb{E}\Big[ \sum_{t=1}^{T} \langle q_t - \overline{q}_t; \ell_t \rangle \Big] \leqslant 2\eta L$$

*Proof.*

$$\eta \overline{\mathcal{R}}_T^R = \eta \sum_{t=1}^{T} \langle q_t - \overline{q}_t; \ell_t \rangle = \alpha \eta \sum_{t=1}^{T} \langle q_t - u; \ell_t \rangle$$
$$\leqslant \alpha \eta \sum_{t=1}^{T} \| q_t - u \|_1$$
$$\leqslant \alpha \eta \sum_{t=1}^{T} \sum_{k=0}^{L-1} \| q_t^k - u \|_1$$
$$\leqslant 2\alpha \eta L T = \frac{2}{1+T} \eta L T$$
$$\leqslant 2\eta L$$

Where the first equality comes from the definition of $\overline{q}_t$, the first inequality follows from Holder's and the fact that $\|\ell_t\|_\infty \leqslant 1$. Finally, the last step derives from bounding the 1-norm between distributions by 2 and the definition of $\alpha = \frac{1}{T+1}$. $\qquad\square$

**Lemma B.4** (Descent Regret Bound). *For $\alpha = \frac{1}{T+1}$, $\eta > 0$ the following fact holds,*

$$\eta \overline{\mathcal{R}}_T^D = \eta \sum_{t=1}^{T} \langle \overline{q}_t - q_t^\circ; \ell_t \rangle \leqslant \eta^2 T L + 18 L^2 \log\left(2X^2 A T\right) \log(T)$$

*Proof.* We start by decomposing the regret at a generic step $t$ into

$$\eta \langle \overline{q}_t - q_t^\circ; \ell_t \rangle = \underbrace{\eta \langle \overline{q}_t - \tilde{q}_{t+1}; \ell_t \rangle}_{\text{Lemma B.5}} + \underbrace{\eta \langle \tilde{q}_{t+1} - q_t^\circ; \ell_t \rangle}_{\text{Lemma B.6}}$$

Where the $\tilde{q}_{t+1}$ represents the unconstrained solution to the OMD optimization problem.

$$q_{t+1} = \underset{q \in \Delta(\overline{P}_t)}{\arg\min}\, D_\psi(q, \tilde{q}_{t+1}), \ \tilde{q}_{t+1} = \underset{q}{\arg\min}\, \langle q; \ell_t \rangle + \frac{1}{\eta} D_\psi(q, \overline{q}_t)$$

We apply Lemma B.5 on $\eta\langle \overline{q}_t - \tilde{q}_{t+1}; \ell_t \rangle$ and Lemma B.6 on $\eta\langle \tilde{q}_{t+1} - q_t^\circ; \ell_t \rangle$ obtaining:

$$\eta\langle \overline{q}_t - q_t^\circ; \ell_t \rangle = D_\psi(\overline{q}_t, \tilde{q}_{t+1}) + D_\psi(\tilde{q}_{t+1}, \overline{q}_t) + D_\psi(q_t^\circ, \overline{q}_t) - D_\psi(\tilde{q}_{t+1}, \overline{q}_t) - D_\psi(q_t^\circ, \tilde{q}_{t+1})$$
$$= D_\psi(\overline{q}_t, \tilde{q}_{t+1}) + D_\psi(q_t^\circ, \overline{q}_t) - D_\psi(q_t^\circ, \tilde{q}_{t+1})$$

Since only the optimality condition for the unconstrained problem and the three-point lemma have been employed so far, this result holds for any $q_t^\circ, \overline{q}_t$. Now, considering the result of the projection step of OMD, $q_{t+1}$, the generalized Pythagorean Theorem allow us to state that $D_\psi(q_t^\circ, \tilde{q}_{t+1}) \geqslant D_\psi(q_t^\circ, q_{t+1})$. Such a result, together with the application of Lemma B.7 to $D_\psi(q_t^\circ, q_{t+1})$ provide:

$$\eta\langle \overline{q}_t - q_t^\circ; \ell_t \rangle = D_\psi(\overline{q}_t, \tilde{q}_{t+1}) + D_\psi(q_t^\circ, \overline{q}_t) - D_\psi(q_t^\circ, \tilde{q}_{t+1})$$
$$\leqslant D_\psi(\overline{q}_t, \tilde{q}_{t+1}) + D_\psi(q_t^\circ, \overline{q}_t) - D_\psi(q_t^\circ, q_{t+1})$$
$$\leqslant D_\psi(\overline{q}_t, \tilde{q}_{t+1}) + D_\psi(q_t^\circ, \overline{q}_t) - \underbrace{D_\psi(q_t^\circ, \tilde{q}_{t+1}) + \frac{\alpha}{(1-\alpha)} D_\psi(q_t^\circ, u)}_{\text{Lemma B.7}}$$

Now, summing over $t \in [\![T]\!]$ we get

$$\eta\overline{\mathcal{R}}_T^D = \eta\sum_{t=1}^T \langle \overline{q}_t - q_t^\circ; \ell_t \rangle \leqslant \underbrace{\sum_{t=1}^T D_\psi(\overline{q}_t, \tilde{q}_{t+1})}_{\substack{\text{``Stability''term} \\ \text{Lemma B.8}}} + \underbrace{\frac{\alpha}{(1-\alpha)} \sum_{t=1}^T D_\psi(q_t^\circ, u)}_{\substack{\text{``Residual'' term} \\ \text{Lemma B.9}}}$$

$$+ \underbrace{\sum_{t=1}^T \left( D_\psi(q_t^\circ, \overline{q}_t) - D_\psi(q_t^\circ, \overline{q}_{t+1}) \right)}_{\substack{\text{``Penalty'' term} \\ \text{Lemma B.10}}}.$$

Now we combine the results from Lemma B.8 for the "Stability" term, Lemma B.9 for the "Residual" term and Lemma B.10 for the "Penalty" term, obtaining:

$$\eta\overline{\mathcal{R}}_T^D \leqslant \eta^2 T L + L\log\left(X^2 A\right) + 17L^2\log\left(2X^2 A T\right)\log(T)$$

$\square$

## B.2 Auxiliary Lemmas

**Lemma B.5.** *For $\overline{q}_t \in \Delta$ and $\tilde{q}_{t+1} = \arg\min_q \langle q; \ell_t \rangle + \frac{1}{\eta}D_\psi(q, \overline{q}_t)$, we have that,*

$$\eta\langle \overline{q}_t - \tilde{q}_{t+1}; \ell_t \rangle = D_\psi(\overline{q}_t, \tilde{q}_{t+1}) + D_\psi(\tilde{q}_{t+1}, \overline{q}_t)$$

*Proof.* Applying the first order optimality conditions for unconstrauned optimisation problems,

$$\langle \overline{q}_t - \tilde{q}_{t+1}; \ell_t + \frac{1}{\eta}(\nabla\psi(\tilde{q}_{t+1}) - \nabla\psi(\overline{q}_t)) \rangle = 0$$

Rearranging and applying the three-point inequality (Chen & Teboulle, 1993),

$$\eta\langle \tilde{q}_{t+1} - \overline{q}_t; \ell_t \rangle = \langle \overline{q}_t - \tilde{q}_{t+1}; \nabla\psi(\tilde{q}_{t+1}) - \nabla\psi(\overline{q}_t) \rangle$$
$$= D_\psi(\overline{q}_t, \tilde{q}_{t+1}) + D_\psi(\tilde{q}_{t+1}, \overline{q}_t).$$

$\square$

**Lemma B.6.** *For $q_t^\circ \in \Delta$ and $\tilde{q}_{t+1} = \arg\min_q \langle q; \ell_t \rangle + \frac{1}{\eta}D_\psi(q, \overline{q}_t)$ we have that,*

$$\eta\langle \tilde{q}_{t+1} - q_t^\circ; \ell_t \rangle = D_\psi(q_t^\circ, \overline{q}_t) - D_\psi(\tilde{q}_{t+1}, \overline{q}_t) - D_\psi(q_t^\circ, \tilde{q}_{t+1})$$

*Proof.* We apply first-order optimality conditions

$$\langle q_t^\circ - \tilde{q}_{t+1}; \ell_t + \frac{1}{\eta}(\nabla\psi(\tilde{q}_{t+1}) - \nabla\psi(\overline{q}_t)) \rangle = 0$$

Rearranging and applying the three-point inequality (Chen & Teboulle, 1993),

$$\eta\langle\tilde{q}_{t+1} - q_t^{\circ}; \ell_t\rangle = \langle q_t^{\circ} - \tilde{q}_{t+1}; \nabla\psi(\tilde{q}_{t+1}) - \nabla\psi(\overline{q}_t)\rangle$$
$$= D_\psi(q_t^{\circ}, \overline{q}_t) - D_\psi(\tilde{q}_{t+1}, \overline{q}_t) - D_\psi(q_t^{\circ}, \tilde{q}_{t+1})$$

$\square$

**Lemma B.7.** *For $\psi\colon \Delta \to \mathbb{R}$ as the negative shannon entropy and for $q_t^{\circ}, q_{t+1} \in \Delta$ and $\overline{q}_{t+1} = (1 - \alpha)q_{t+1} + \alpha u$ we have:*

$$D_\psi(q_t^{\circ}, q_{t+1}) \geqslant D_\psi(q_t^{\circ}, \overline{q}_{t+1}) - \frac{\alpha}{(1 - \alpha)} D_\psi(q_t^{\circ}, u).$$

*Proof.* Due to the convexity of the regularizer, namely the KL divergence, we observe that

$$D_\psi(q_t^{\circ}, \overline{q}_{t+1}) = D_\psi(q_t^{\circ}, (1 - \alpha)q_{t+1} + \alpha u) \leqslant (1 - \alpha)D_\psi(q_t^{\circ}, q_{t+1}) + \alpha D_\psi(q_t^{\circ}, u)$$
$$\frac{1}{(1 - \alpha)} D_\psi(q_t^{\circ}, \overline{q}_{t+1}) - \frac{\alpha}{(1 - \alpha)} D_\psi(q_t^{\circ}, u) \leqslant D_\psi(q_t^{\circ}, q_{t+1})$$

We retrieve that:

$$D_\psi(q_t^{\circ}, q_{t+1}) \geqslant D_\psi(q_t^{\circ}, \overline{q}_{t+1}) - \frac{\alpha}{(1 - \alpha)} D_\psi(q_t^{\circ}, u)$$

Where the last step comes from the fact that $1 - \alpha \leqslant 1$. $\square$

**Lemma B.8** ("Stability" Term Bound). *For $\eta > 0$, $\ell_t\colon \mathcal{X} \times \mathcal{A} \to [0, 1]$, $\overline{q}_t \in \Delta$ and $\tilde{q}_{t+1} = \arg\min_q\langle q; \ell_t\rangle + \frac{1}{\eta}D_\psi(q, \overline{q}_t)$ we have that:*

$$\textit{(STAB)} \quad \sum_{t=1}^{T} D_\psi(\overline{q}_t, \tilde{q}_{t+1}) \leqslant \eta^2 TL.$$

*Proof.* The term can be easily bounded following standard analysis of OMD:

$$\sum_{t=1}^{T} D_\psi(\overline{q}_t, \tilde{q}_{t+1})$$

$$= \sum_{t=1}^{T} \Bigg[ \sum_{x,a,x'} \overline{q}_t(x, a, x') \log\left(\frac{\overline{q}_t(x, a, x')}{\tilde{q}_{t+1}(x, a, x')}\right) - \sum_{x,a,x'} \left(\overline{q}_t(x, a, x') - \tilde{q}_{t+1}(x, a, x')\right) \Bigg]$$

$$= \sum_{t=1}^{T} \Bigg[ \sum_{x,a,x'} \eta\ell_t(x, a)\overline{q}_t(x, a, x') + \overline{q}_t(x, a, x') \exp\left(-\eta\ell_t(x, a)\right) - \overline{q}_t(x, a, x') \Bigg]$$

$$\leqslant \sum_{t=1}^{T} \Bigg[ \sum_{(x,a,x')} \eta^2 \overline{q}_t(x, a, x')\ell_t^2(x, a) \Bigg]$$

$$\leqslant \eta^2 TL$$

Where the first equality comes from the definition of generalized KL divergence, the second by applying the definition of the solution of the unconstrained optimization problem, namely:

$$\tilde{q}_{t+1}(x, a, x') = \overline{q}_t(x, a, x') \exp\left(-\eta\ell_t(x, a)\right), \forall(x, a, x') \in \mathcal{X} \times \mathcal{A} \times \mathcal{X}$$

and further simplifications. Finally, we use the standard bound of the exponential function $e^{-x} \leqslant 1 - x - x^2, \forall x \geqslant 0$, the fact that losses are bounded, $\ell_t(x, a) \in [0, 1]$ and as a last step we use the fundamental property of occupancies $\sum_{(x,a,x')\in\mathcal{X}\times\mathcal{A}\times\mathcal{X}} \overline{q}_t(x, a, x') = L$. $\square$

**Lemma B.9** ("Residual" Term Bound). *For $\alpha = \frac{1}{T+1}$, $u(x, a, x') = \frac{1}{X_k A X_{k+1}}$ and $q_t^{\circ} \in \Delta$ it holds that*

$$\textit{(RES)} \quad \frac{\alpha}{1 - \alpha} \sum_{t=1}^{T} D_\psi(q_t^{\circ}, u) \leqslant L \log\left(X^2 A\right)$$

*Proof.* We first notice that

$$D_\psi(q_t^\circ, u) = \sum_{k=0}^{L-1} \sum_{x \in \mathcal{X}_k} \sum_{a \in \mathcal{A}} \sum_{x' \in \mathcal{X}_{k+1}} q_t^\circ(x, a, x') \log \left( \frac{q_t^\circ(x, a, x')}{\frac{1}{X_k A X_{k+1}}} \right)$$

$$\leqslant \sum_{k=0}^{L-1} \sum_{x \in \mathcal{X}_k} \sum_{a \in \mathcal{A}} \sum_{x' \in \mathcal{X}_{k+1}} q_t^\circ(x, a, x') \log \left( X^2 A \right)$$

$$\leqslant L \log \left( X^2 A \right)$$

The first inequality comes from neglecting negative terms and the last using the fundamental property of occupancy measures.

Thus, considering the sum over $t$:

$$\frac{\alpha}{1-\alpha} \sum_{t=1}^{T} D_\psi(q_t^\circ, u) \leqslant \frac{\alpha}{1-\alpha} T L \log \left( X^2 A \right)$$

And chosing $\alpha$ so that $\frac{\alpha}{1-\alpha} T = 1$, namely $\alpha = \frac{1}{T+1}$ leads to the desired bound. $\qquad \square$

**Lemma B.10** ("Penalty" Term Bound). *For $\overline{q}_t = (1-\alpha)q_t + \alpha u$ and for $\alpha = \frac{1}{T+1}$*

$$(PEN) \quad \sum_{t=1}^{T} (D_\psi(q_t^\circ, \overline{q}_t) - D_\psi(q_t^\circ, \overline{q}_{t+1})) \leqslant 17 L^2 \log \left( 2X^2 A T \right) \log(T)$$

*Proof.* First, we unravel the summation:

$$\sum_{t=1}^{T} (D_\psi(q_t^\circ, \overline{q}_t) - D_\psi(q_t^\circ, \overline{q}_{t+1})) = D_\psi(q_1^\circ, \overline{q}_1) - D_\psi(q_1^\circ, \overline{q}_2) + D_\psi(q_T^\circ, \overline{q}_T) - D_\psi(q_T^\circ, \overline{q}_{T+1})$$

$$\leqslant \underbrace{D_\psi(q_1^\circ, \overline{q}_1)}_{\substack{\text{Range of } \psi \\ \text{Lemma B.11}}} + \underbrace{\sum_{t=2}^{T} D_\psi(q_t^\circ, \overline{q}_t) - D_\psi(q_{t-1}^\circ, \overline{q}_t)}_{\substack{\text{Non-telescoping term} \\ \text{Lemma B.12}}}$$

Where the inequality comes from neglecting negative terms. What is left to do is to combine Lemma B.11, that expresses the range of the negative shannon entropy, and Lemma B.12 providing a bound for the non-telescoping term.

$$\sum_{t=1}^{T} (D_\psi(q_t^\circ, \overline{q}_t) - D_\psi(q_t^\circ, \overline{q}_{t+1})) \leqslant L \log \left( X^2 A \right) + 16 L^2 \log(2X^2 A T) \log(T)$$

$\qquad \square$

**Lemma B.11** (Range of $\psi$ Bound). *For $q_t^\circ \in \Delta$ and $\overline{q}_1 = (1-\alpha)q_1 + \alpha u$ we have that,*

$$D_\psi(q_1^\circ, \overline{q}_1) \leqslant L \log \left( X^2 A \right)$$

*Proof.* The steps are similar to the ones of Lemma B.9:

$$D_\psi(q_1^\circ, \overline{q}_1) = D_\psi(q_1^\circ, u) = \sum_{k=0}^{L-1} \sum_{x \in \mathcal{X}_k} \sum_{a \in \mathcal{A}} \sum_{x' \in \mathcal{X}_{k+1}} q_1^\circ(x, a, x') \log \left( \frac{q_1^\circ(x, a, x')}{\frac{1}{X_k A X_{k+1}}} \right)$$

$$\leqslant \sum_{k=0}^{L-1} \sum_{x \in \mathcal{X}_k} \sum_{a \in \mathcal{A}} \sum_{x' \in \mathcal{X}_{k+1}} q_1^\circ(x, a, x') \log \left( X^2 A \right)$$

$$\leqslant L \log \left( X^2 A \right).$$

$\qquad \square$

**Lemma B.12** (Non-telescoping term Bound). *For $q_t^\circ \in \Delta$ and $\overline{q}_t = (1-\alpha)q_t + \alpha u$ we have that:*

$$\sum_{t=2}^{T} D_\psi(q_t^\circ, \overline{q}_t) - D_\psi(q_{t-1}^\circ, \overline{q}_t) \leqslant 16L^2 \log(2X^2 AT) \log(T).$$

*Proof.* We first focus on each time-step $t$. From the definition of the Bregman Divergence we have:

$$\begin{aligned}
D_\psi(q_t^\circ, \overline{q}_t) - D_\psi(q_{t-1}^\circ, \overline{q}_t) &= \psi(q_t^\circ) - \psi(q_{t-1}^\circ) + \langle \nabla \psi(\overline{q}_t); q_{t-1}^\circ - q_t^\circ \rangle \\
&= \psi(q_t^\circ) - \psi(q_{t-1}^\circ) + \langle \log(\overline{q}_t); q_{t-1}^\circ - q_t^\circ \rangle + \langle \mathbf{1}; q_{t-1}^\circ - q_t^\circ \rangle \\
&\leqslant |\psi(q_t^\circ) - \psi(q_{t-1}^\circ)| + \|\log(\overline{q}_t)\|_\infty \|q_t^\circ - q_{t-1}^\circ\|_1
\end{aligned}$$

Where the second equality comes from the computation of the derivative of the Negative Shannon Entropy:

$$\nabla \psi(f) = \left( \frac{\partial \psi(f)}{\partial f} \right)^T = (\log(f) + 1, \dots)^T$$

Finally, the last step comes from Holder's inequality and from taking the absolute value of the difference of the entropies.

We combine the results from Lemma B.13 and Lemma B.14 to obtain over the whole time horizon:

$$\begin{aligned}
\sum_{t=2}^{T} D_\psi(q_t^\circ, \overline{q}_t) - D_\psi(q_{t-1}^\circ, \overline{q}_t) &\leqslant \underbrace{\sum_{t=2}^{T} |\psi(q_t^\circ) - \psi(q_{t-1}^\circ)|}_{\text{Lemma B.13}} + \underbrace{\sum_{t=2}^{T} \|\log(\overline{q}_t)\|_\infty \|q_t^\circ - q_{t-1}^\circ\|_1}_{\text{Lemma B.14}} \\
&\leqslant 8L^2 \log\left(2X^2 AT\right) \log(T) + 8L^2 \log\left(X^2 AT + X^2 A\right) \log(T)
\end{aligned}$$

$\square$

**Lemma B.13.** *it holds that*

$$\sum_{t=2}^{T} \left| \psi\left(q_t^\circ\right) - \psi\left(q_{t-1}^\circ\right) \right| \leqslant 8L^2 \log\left(2X^2 AT\right) \log(T)$$

*Proof.* The proof comes from applying Lemma B.15 and Lemma B.16 to the following decomposition:

$$\begin{aligned}
\sum_{t=2}^{T} \left| \psi\left(q_t^\circ\right) - \psi\left(q_{t-1}^\circ\right) \right| &= \underbrace{\sum_{t=2}^{\underline{t}} \left| \psi\left(q_t^\circ\right) - \psi\left(q_{t-1}^\circ\right) \right|}_{\text{Lemma B.15}} + \underbrace{\sum_{t=\overline{t}}^{T} \left| \psi\left(q_t^\circ\right) - \psi\left(q_{t-1}^\circ\right) \right|}_{\text{Lemma B.16}} \\
&\leqslant 2L^2 \log\left(X^2 A\right) + 6L^2 \log\left(2X^2 AT\right) \log(T)
\end{aligned}$$

Simplifying the expression with another upper bound completes the proof. $\square$

**Lemma B.14.** *Defined $\overline{t} := \left\lceil \frac{L}{1 - \frac{1}{X^2 A}} \right\rceil$ and $\underline{t} := \left\lfloor \frac{L}{1 - \frac{1}{X^2 A}} \right\rfloor$, for $\overline{q}_t = (1-\alpha)q_t + \alpha u$ and $\alpha = \frac{1}{T+1}$ it holds that,*

$$\sum_{t=2}^{T} \|\log(\overline{q}_t)\|_\infty \left\| q_t^\circ - q_{t-1}^\circ \right\|_1 \leqslant 8L^2 \log\left(X^2 AT + X^2 A\right) \log(T)$$

*Proof.* The proof comes from applying Lemma B.20 and Lemma B.21 to the following decomosition,

$$\sum_{t=2}^{T} \|\log(\overline{q}_t)\|_\infty \left\| q_t^\circ - q_{t-1}^\circ \right\|_1 = \underbrace{\sum_{t=2}^{\underline{t}} \|\log(\overline{q}_t)\|_\infty \left\| q_t^\circ - q_{t-1}^\circ \right\|_1}_{\text{Lemma B.20}} + \underbrace{\sum_{t=\overline{t}}^{T} \|\log(\overline{q}_t)\|_\infty \left\| q_t^\circ - q_{t-1}^\circ \right\|_1}_{\text{Lemma B.21}}.$$

$\square$

**Lemma B.15.** *For $X^2A \geqslant 2$ and $\underline{t} := \left\lfloor \frac{L}{1-\frac{1}{X^2A}} \right\rfloor$*

$$\sum_{t=2}^{\underline{t}} \left| \psi\left(q_t^\circ\right) - \psi\left(q_{t-1}^\circ\right) \right| \leqslant 2L^2 \log(X^2A)$$

*Proof.*

$$\sum_{t=2}^{\underline{t}} \left| \psi\left(q_t^\circ\right) - \psi\left(q_{t-1}^\circ\right) \right| \leqslant \sum_{t=2}^{\underline{t}} \sum_{k=0}^{L-1} \left| \psi\left(q_t^{\circ,k}\right) - \psi\left(q_{t-1}^{\circ,k}\right) \right|$$

$$\leqslant \sum_{t=2}^{\underline{t}} L \log(X^2A)$$

$$\leqslant L \log(X^2A)\underline{t}$$

$$\leqslant 2L^2 \log(X^2A)$$

Where the second inequality comes from Lemma B.17, while the last step comes from observing that, for $X^2A \geqslant 2$,

$$\underline{t} \leqslant \frac{L}{1 - \frac{1}{X^2A}}$$

$$\leqslant L\frac{X^2A}{X^2A - 1}$$

$$\leqslant 2L$$

$\square$

**Lemma B.16.** *For $\bar{t} := \left\lceil \frac{L}{1-\frac{1}{X^2A}} \right\rceil$ and $\epsilon_t = \frac{L}{t}$ it holds that,*

$$\sum_{t=\bar{t}}^{T} \left| \psi\left(q_t^\circ\right) - \psi\left(q_{t-1}^\circ\right) \right| \leqslant 6L^2 \log\left(2X^2AT\right) \log(T)$$

*Proof.*

$$\sum_{t=\bar{t}}^{T} \left| \psi\left(q_t^\circ\right) - \psi\left(q_{t-1}^\circ\right) \right| \leqslant \sum_{t=\bar{t}}^{T} \sum_{k=0}^{L-1} h\left(\epsilon_t\right) + \log\left(X^2AT + X^2A\right)\epsilon_t$$

$$\leqslant 2L^2 \log(T) + L^2 \log^2(T) + L^2$$

$$+ L^2 \log\left(X^2AT + X^2A\right)(\log(T) + 1)$$

Where the first inequality comes from Lemma B.18, the second inequality comes from Lemma B.19 and from the bound on the harmonic series. Further upperbounding to simplify the expression finishes the proof. $\square$

**Lemma B.17** (Per-step entropy difference bound). *For any $t \leqslant \underline{t} := \left\lfloor \frac{L}{1-\frac{1}{X^2A}} \right\rfloor$ it holds that,*

$$\left| \psi\left(q_t^\circ\right) - \psi\left(q_{t-1}^\circ\right) \right| \leqslant L \log\left(X^2A\right)$$

*Proof.*

$$\left| \psi\left(q_t^\circ\right) - \psi\left(q_{t-1}^\circ\right) \right| \leqslant \sum_{k=0}^{L-1} \left| \psi\left(q_t^{\circ,k}\right) - \psi\left(q_{t-1}^{\circ,k}\right) \right|$$

$$\leqslant \sum_{k=0}^{L-1} \log(X_kAX_{k+1})$$

$$\leqslant \sum_{k=0}^{L-1} \log(X^2 A)$$

$$\leqslant L \log(X^2 A)$$

Again, the first inequality comes from triangle inequality, the rest is a direct consequance of (Theorem 3, Sason, 2013). □

**Lemma B.18** (Per-step entropy difference bound). *For any $t \geqslant \bar{t} := \left\lceil \frac{L}{1 - \frac{1}{X^2 A}} \right\rceil$ and $\epsilon_t = \frac{L}{t}$ it holds that,*

$$\left| \psi\left(q_t^\circ\right) - \psi\left(q_{t-1}^\circ\right) \right| \leqslant \sum_{k=0}^{L-1} h\left(\epsilon_t\right) + \log\left(X^2 AT + X^2 A\right) \epsilon_t$$

*Proof.* Denoting with $\epsilon_{t,k} = d_{TV}\left(q_t^{\circ,k}, q_{t-1}^{\circ,k}\right)$ the total variation distance between the occupancies at layer $k$, we notice that for any $t \in [\![T]\!]$.

$$\left| \psi\left(q_t^\circ\right) - \psi\left(q_{t-1}^\circ\right) \right| \leqslant \sum_{k=0}^{L-1} \left| \psi\left(q_t^{\circ,k}\right) - \psi\left(q_{t-1}^{\circ,k}\right) \right|$$

$$\leqslant \sum_{k=0}^{L-1} h\left(\epsilon_{t,k}\right) + \epsilon_{t,k} \log\left(X^2 A - 1\right)$$

$$\leqslant \sum_{k=0}^{L-1} h\left(\epsilon_{t,k}\right) + \epsilon_{t,k} \log\left(X^2 AT + X^2 A\right)$$

$$\leqslant \sum_{k=0}^{L-1} h\left(\epsilon_t\right) + \epsilon_t \log\left(X^2 AT + X^2 A\right)$$

The first inequality comes from triangle inequality. The rest comes from a straightforward application of (Theorem 3, Sason, 2013). □

**Lemma B.19** (Binary Entropy Bound). *For $\bar{t} := \left\lceil \frac{L}{1 - \frac{1}{X^2 A}} \right\rceil$ and $\epsilon_t = \frac{L}{t}$ it holds that,*

$$\sum_{t=\bar{t}}^{T} \sum_{k=0}^{L-1} h\left(\epsilon_t\right) \leqslant 2L^2 \log(T) + L^2 \log^2(T) + L^2$$

*Proof.* For each $t \in [\bar{t}, T]$ we have that,

$$h(\epsilon_t) = \frac{L}{t} \log\left(\frac{t}{L}\right) + \left(1 - \frac{L}{t}\right) \log\left(\frac{t}{t-L}\right)$$

$$\leqslant \frac{L}{t} \log\left(\frac{t}{L}\right) + \log\left(\frac{t}{t-L}\right)$$

$$\leqslant L \log(T) \frac{1}{t} + \log\left(\frac{t}{t-L}\right)$$

$$\leqslant L \log(T) \frac{1}{t} + \frac{L}{t'}.$$

The last inequality makes use of

$$\log\left(\frac{t}{t-L}\right) = \log\left(\frac{t'+L}{t'}\right) = \log\left(1 + \frac{L}{t'}\right) \leqslant \frac{L}{t'}, \quad (t = t' + L)$$

Now considering the summations over $k$ and $t$,

$$\sum_{t=\bar{t}}^{T} \sum_{k=0}^{L-1} h(\epsilon_t) \leqslant L \left( \sum_{t=\bar{t}}^{T} L \log(T) \frac{1}{t} + \frac{L}{t'} \right)$$

$$\leqslant L^2 \log(T) \sum_{t=1}^{T} \frac{1}{t} + L^2 \sum_{t'=1}^{T-L} \frac{1}{t'}$$

$$\leqslant 2L^2 \log(T) + L^2 \log^2(T) + L^2$$

where the last inequality comes from the fact that $\sum_{t'=1}^{T-L} \frac{1}{t'} \leqslant 1 + \log(T-L) \leqslant 1 + \log(T)$. □

**Lemma B.20.** *For $t \in [\![2, \bar{t}]\!]$, where $\bar{t} := \left\lceil \frac{L}{1 - \frac{1}{X^2 A}} \right\rceil$, it holds that,*

$$\sum_{t=2}^{\bar{t}} \|\log(\bar{q}_t)\|_\infty \|q_t^\circ - q_{t-1}^\circ\|_1 \leqslant 4L^2 \log\left(X^2 AT + X^2 A\right) \log(T)$$

*Proof.*

$$\sum_{t=2}^{\bar{t}} \|\log(\bar{q}_t)\|_\infty \|q_t^\circ - q_{t-1}^\circ\|_1 \leqslant 2 \sum_{t=2}^{\bar{t}} \sum_{k=0}^{L-1} \log\left(X^2 AT + X^2 A\right) \epsilon_{t,k}$$

$$\leqslant 2 \sum_{t=2}^{\bar{t}} \sum_{k=0}^{L-1} \log\left(X^2 AT + X^2 A\right) \epsilon_t$$

$$\leqslant 2 \sum_{t=2}^{\bar{t}} \sum_{k=0}^{L-1} \log\left(X^2 AT + X^2 A\right) \frac{L}{t}$$

$$= 2L^2 \log\left(X^2 AT + X^2 A\right) \sum_{t=2}^{\bar{t}} \frac{1}{t}$$

$$\leqslant 2L^2 \log\left(X^2 AT + X^2 A\right) (\log(\underline{t}) + 1)$$

$$\leqslant 2L^2 \log\left(X^2 AT + X^2 A\right) (\log(T) + 1)$$

Where the first inequality comes from Lemma B.22, the second from the bound Lemma F.4 and the last step comes from the monotonicity of the logarithm. □

**Lemma B.21.** *For $\bar{t} := \left\lceil \frac{L}{1 - \frac{1}{X^2 A}} \right\rceil$, $\epsilon_{t,k} := d_{TV}(q_t^{\circ,k}, q_{t-1}^{\circ,k})$ and $\epsilon_t = \frac{L}{t}$ it holds that,*

$$\sum_{t=\bar{t}}^{T} \|\log(\bar{q}_t)\|_\infty \|q_t^\circ - q_{t-1}^\circ\|_1 \leqslant 4L^2 \log\left(X^2 AT + X^2 A\right) \log(T)$$

*Proof.*

$$\sum_{t=\bar{t}}^{T} \|\log(\bar{q}_t)\|_\infty \|q_t^\circ - q_{t-1}^\circ\|_1 \leqslant 2 \sum_{t=\underline{t}}^{T} \sum_{k=0}^{L-1} \log\left(X^2 AT + X^2 A\right) \epsilon_{t,k}$$

$$\leqslant 2 \sum_{t=\underline{t}}^{T} \sum_{k=0}^{L-1} \log\left(X^2 AT + X^2 A\right) \epsilon_t$$

$$\leqslant 2L^2 \log\left(X^2 AT + X^2 A\right) (\log(T) + 1)$$

Where the first inequality comes from Lemma B.22, the second from the bound Lemma F.4. The last step comes from the bound on the harmonic series. □

**Lemma B.22.** *For any $t \in [\![T]\!]$ and $\epsilon_{t,k} = d_{TV}\left(q_t^{\circ,k}, q_{t-1}^{\circ,k}\right)$ it holds that:*

$$\|\log(\bar{q}_t)\|_\infty \|q_t^\circ - q_{t-1}^\circ\|_1 \leqslant 2 \sum_{k=0}^{L-1} \log\left(X^2 AT + X^2 A\right) \epsilon_{t,k}$$

*Proof.* First, we notice that for any $t \in [\![T]\!]$:

$$
\begin{aligned}
\|\log(\overline{q}_t)\|_\infty &= \left\| \left( \log(\overline{q}_t(x, a, x')), \ldots \right)^\top \right\|_\infty \\
&\leqslant \left\| \left( \log(\alpha u(x, a, x')), \ldots \right)^\top \right\|_\infty \\
&= \left\| \left( \log\left( \alpha \frac{1}{X^2 A X} \right), \ldots \right)^\top \right\|_\infty \\
&= \log\left( \frac{X^2 A}{\alpha} \right) \\
&= \log\left( X^2 A T + X^2 A \right)
\end{aligned}
$$

Where the first inequality comes from the definition $\overline{q}_t(x, a, x') := (1 - \alpha) q_t(x, a, x') + \alpha u(x, a, x'), \forall (x, a, x') \in \mathcal{X} \times \mathcal{A} \times \mathcal{X}$ and the monotonicity of the logarithm.

Following this result, it holds that for any $t \in [\![T]\!]$:

$$
\begin{aligned}
\|\log(\overline{q}_t)\|_\infty \left\| q_t^\circ - q_{t-1}^\circ \right\|_1 &= \sum_{k=0}^{L-1} \log\left( X^2 A T + X^2 A \right) \left\| q_t^{\circ,k} - q_{t-1}^{\circ,k} \right\|_1 \\
&= 2 \sum_{k=0}^{L-1} \log\left( X^2 A T + X^2 A \right) d_{TV}\left( q_t^{\circ,k}, q_{t-1}^{\circ,k} \right) \\
&= 2 \sum_{k=0}^{L-1} \log\left( X^2 A T + X^2 A \right) \epsilon_{t,k}
\end{aligned}
$$

$\square$

# C THEORETICAL ANALYSIS WITH FULL-FEEDBACK AND GENERIC SMOOTHING FUNCTIONS

In this section, we present the proofs of the results discussed in Section 4 related to the regret analysis agnostic of the smoothing function to be used. For convenience, we will restate the Theorems and Lemmas before providing a detailed analysis of each and report just the Lemmas that differentiate from Appendix B.

## C.1 MAIN RESULTS

For the proposed Algorithm, we can provide the following:

**Theorem 4.1** (Smoothed-Regret Bound for SOMD under full-feedback). *Let* $\eta = \sqrt{(10 L \log(2 X^2 A T) \rho_T^f)/T}$ *and* $\alpha = 1/(1+T)$, *then for any comparator policy* $\pi^\circ \in \Pi$ *Algorithm 1 suffers a smoothed regret of:*

$$
\overline{\mathcal{R}}_T(\pi^\circ) \leqslant \mathcal{O}\left( L^2 \overline{D}_f^{\mathsf{P}} + L^{3/2} \sqrt{T \log(X^2 A T) \rho_T^f} \right), \tag{11}
$$

*where* $\rho_T^f := \log(T) + \overline{D}_f^{\mathsf{P}} + \mathcal{H}_T(\overline{D}_f^{\mathsf{P}})$ *and* $\mathcal{H}_T(\overline{D}_f^{\mathsf{P}}) = T L h\left( \frac{(L^2 + L) \overline{D}_f^{\mathsf{P}}}{2 T L} \right)$.

*Proof.* We first define $q_t^{\overline{P}_{t-1}, \pi_t} = q_t$ and $q^{\overline{P}_t, \pi^\circ} = q_t^\circ$ for notational simplicity. It follows that the smoothed regret term can be decomposed as

$$
\overline{\mathcal{R}}_T(\pi^\circ) = \mathbb{E}\left[ \sum_{t=1}^T \langle q^{\overline{P}_t, \pi_t} - q^{\overline{P}_t, \pi^\circ}; \ell_t \rangle \right] = \underbrace{\mathbb{E}\left[ \sum_{t=1}^T \langle q_t - q_t^\circ; \ell_t \rangle \right]}_{\substack{\text{Algorithmic Regret } (\overline{\mathcal{R}}_T^A) \\ \text{Lemma C.3}}} + \underbrace{\mathbb{E}\left[ \sum_{t=1}^T \langle q^{\overline{P}_t, \pi_t} - q^{\overline{P}_{t-1}, \pi_t}; \ell_t \rangle \right]}_{\substack{\text{Update Regret } (\overline{\mathcal{R}}_T^U) \\ \text{Lemma C.2}}},
$$

Then, the result follows directly from the combination of Lemma C.2 for the Update Regret and Lemma C.3 for the Algorithmic Regret, namely:

$$\overline{\mathcal{R}}_T(\pi^\circ) \leqslant (L^2 + L)\overline{D}_f^{\mathsf{P}} + 2L + 2L\sqrt{10LT\log(2X^2AT)\rho_T^f}$$

which leads to the final result. $\qquad\qquad\qquad\qquad\qquad\qquad\qquad\qquad\square$

**Lemma C.1** (Trend of $\mathcal{H}_T(\overline{D}_f^{\mathsf{P}})$ for average smoothing functions). *Let* $\overline{P}_t := \frac{1}{t}\sum_{t'\in[\![t]\!]} P_{t'}$, *then* $\overline{D}_f^{\mathsf{P}} \approx L\log T$ *and*

$$\mathcal{H}_T(\overline{D}_f^{\mathsf{P}}) = TLh\left(\frac{(L^2+L)\overline{D}_f^{\mathsf{P}}}{2TL}\right)$$

$$= TL\frac{(L^2+L)\log T}{2T}\log\frac{2T}{(L^2+L)\log T} +$$

$$+ TL\left(1 - \frac{(L^2+L)\log T}{2T}\right)\log\left(\frac{2T}{2T-(L^2+L)\log T}\right)$$

$$\approx L^2(\log T)^2.$$

**Lemma C.2** (Update Regret Bound). *For* $\ell_t : \mathcal{X} \times \mathcal{A} \to [0,1]$, *for any policy sequence* $\{\pi_t\}_{t\in[\![T]\!]}$ *and defined* $\overline{D}_f^{\mathsf{P}} = \sum_{t=2}^{T}\max_{(x,a)\in\mathcal{X}\times\mathcal{A}}\|\overline{P}_t(\cdot|x,a) - \overline{P}_{t-1}(\cdot|x,a)\|_1$, *then for a generic smoothing function* $f$, *we have that,*

$$\overline{\mathcal{R}}_T^U = \mathbb{E}\Big[\sum_{t=1}^{T}\langle q^{\overline{P}_t,\pi_t} - q^{\overline{P}_{t-1},\pi_t}; \ell_t\rangle\Big] \leqslant (L^2+L)\overline{D}_f^{\mathsf{P}}$$

*Proof.*

$$\overline{\mathcal{R}}_T^U = \mathbb{E}\Big[\sum_{t=1}^{T}\langle q^{\overline{P}_t,\pi_t} - q^{\overline{P}_{t-1},\pi_t}; \ell_t\rangle\Big] \leqslant \mathbb{E}\Big[\sum_{t=1}^{T}\Big\|q^{\overline{P}_t,\pi_t} - q^{\overline{P}_{t-1},\pi_t}\Big\|_1\Big]$$

$$\leqslant (L^2+L)\sum_{t=1}^{T}\max_{(x,a)\in\mathcal{X}\times\mathcal{A}}\|\overline{P}_t(\cdot|x,a) - \overline{P}_{t-1}(\cdot|x,a)\|_1$$

$$\leqslant (L^2+L)\overline{D}_f^{\mathsf{P}}$$

The first inequality follows from Holder's and the fact that $\|\ell_t\|_\infty \leqslant 1$. The second inequality comes from Lemma F.10 and the remaining step follow from the definition of $\overline{D}_f^{\mathsf{P}}$. $\qquad\square$

**Lemma C.3** (Algorithmic Regret Bound). *Choosing* $\eta = \sqrt{\frac{10L\log(2X^2AT)\rho_T^f}{T}}$, $\alpha = \frac{1}{1+T}$ , $\epsilon_{t,k} = d_{TV}(q_t^{\circ,k}, q_{t-1}^{\circ,k})$, $\mathcal{H}_T(\overline{D}_f^{\mathsf{P}}) = TLh\left(\frac{(L^2+L)\overline{D}_f^{\mathsf{P}}}{2TL}\right)$, $\overline{D}_f^{\mathsf{P}} = \sum_{t=2}^{T}\max_{(x,a)\in\mathcal{X}\times\mathcal{A}}\|\overline{P}_t(\cdot|x,a) - \overline{P}_{t-1}(\cdot|x,a)\|_1$ *and for* $\rho_T^f = \log(T) + \overline{D}_f^{\mathsf{P}} + \mathcal{H}_T(\overline{D}_f^{\mathsf{P}})$, *the Algorithmic Regret is bounded by*

$$\overline{\mathcal{R}}_T^A = \mathbb{E}\Big[\sum_{t=1}^{T}\langle q_t - q_t^\circ; \ell_t\rangle\Big] \leqslant 2L + 2L\sqrt{10LT\log(2X^2AT)\rho_T^f} \qquad (20)$$

*Proof.* The proof relies on the following decomposition:

$$\eta\overline{\mathcal{R}}_T^A = \eta\mathbb{E}\Big[\sum_{t=1}^{T}\langle q_t - q_t^\circ; \ell_t\rangle \pm \eta\sum_{t=1}^{T}\langle \overline{q}_t; \ell_t\rangle\Big] = \eta\underbrace{\mathbb{E}\Big[\sum_{t=1}^{T}\langle \overline{q}_t - q_t^\circ; \ell_t\rangle\Big]}_{\substack{\text{Descent Regret } (\overline{\mathcal{R}}_T^D) \\ \text{Lemma C.4}}} + \eta\underbrace{\mathbb{E}\Big[\sum_{t=1}^{T}\langle q_t - \overline{q}_t; \ell_t\rangle\Big]}_{\substack{\text{Regularization Regret } (\overline{\mathcal{R}}_T^R) \\ \text{Lemma B.3}}}.$$

By applying Lemma B.3 for the Regularization Regret and Lemma C.4 for the Descent Regret we can rewrite the Algorithmic Regret as

$$\eta \overline{\mathcal{R}}_T^A \leqslant 2\eta L + \eta^2 TL + \mathcal{H}_T(\overline{D}_f^{\mathsf{P}}) + L^2 \log(2X^2 AT)\overline{D}_f^{\mathsf{P}} + 10L^2 \log\left(2X^2 AT\right) \log(T)$$

$$\leqslant 2\eta L + \eta^2 TL + 10(L^2 \log\left(2X^2 AT\right)\left(\log(T) + \overline{D}_f^{\mathsf{P}} + \mathcal{H}_T(\overline{D}_f^{\mathsf{P}}))\right.$$

$$= 2\eta L + \eta^2 TL + 10L^2 \log\left(2X^2 AT\right) \rho_T^f$$

where $\rho_T^f = \log(T) + \overline{D}_f^{\mathsf{P}} + \mathcal{H}_T(\overline{D}_f^{\mathsf{P}})$. Choosing the optimal $\eta$:

$$\eta = \sqrt{\frac{10L \log(2X^2 AT)\rho_T^f}{T}}$$

and then we upper-bound the Algorithmic Regret after substituting the optimal $\eta$, namely:

$$\overline{\mathcal{R}}_T^A \leqslant 2L + \eta TL + \frac{10L^2 \log\left(2X^2 AT\right) \rho_T^f}{\eta}$$

$$\leqslant 2L + L\sqrt{10LT \log(2X^2 AT)\rho_T^f} + L\sqrt{10LT \log(2X^2 AT)\rho_T^f}$$

$$\leqslant 2L + 2L\sqrt{10LT \log(2X^2 AT)\rho_T^f}$$

which leads to the final result. $\qquad\square$

**Lemma C.4** (Descent Regret Bound). *For* $\alpha = \frac{1}{T+1}$ *,* $\epsilon_{t,k} = d_{TV}(q_t^{\circ,k}, q_{t-1}^{\circ,k})$, $\mathcal{H}_T(\overline{D}_f^{\mathsf{P}}) = TLh\left(\frac{(L^2+L)\overline{D}_f^{\mathsf{P}}}{2TL}\right)$ *and* $\overline{D}_f^{\mathsf{P}} = \sum_{t=2}^T \max_{(x,a)\in\mathcal{X}\times\mathcal{A}} \|\overline{P}_t(\cdot|x,a) - \overline{P}_{t-1}(\cdot|x,a)\|_1$ *the following fact holds*

$$\eta\overline{\mathcal{R}}_T^D = \eta \sum_{t=1}^T \langle \overline{q}_t - q_t^\circ; \ell_t\rangle \leqslant \eta^2 TL + \mathcal{H}_T(\overline{D}_f^{\mathsf{P}}) + L^2 \log(2X^2 AT)\overline{D}_f^{\mathsf{P}} + 10L^2 \log\left(2X^2 AT\right) \log(T)$$

*Proof.* We start by decomposing the regret at a generic step $t$ into

$$\eta\langle \overline{q}_t - q_t^\circ; \ell_t\rangle = \eta\langle \overline{q}_t - \tilde{q}_{t+1}; \ell_t\rangle + \eta\langle \tilde{q}_{t+1} - q_t^\circ; \ell_t\rangle$$

Where the $\tilde{q}_{t+1}$ represents the unconstrained solution to the OMD optimization problem.

$$q_{t+1} = \arg\min_{q\in\Delta(\overline{P}_t)} D_\psi(q, \tilde{q}_{t+1}), \ \tilde{q}_{t+1} = \arg\min_q \langle q; \ell_t\rangle + \frac{1}{\eta}D_\psi(q, \overline{q}_t)$$

We apply Lemma B.5 on $\eta\langle \overline{q}_t - \tilde{q}_{t+1}; \ell_t\rangle$ and Lemma B.6 on $\eta\langle \tilde{q}_{t+1} - q_t^\circ; \ell_t\rangle$ obtaining:

$$\eta\langle \overline{q}_t - q_t^\circ; \ell_t\rangle = D_\psi(\overline{q}_t, \tilde{q}_{t+1}) + D_\psi(\tilde{q}_{t+1}, \overline{q}_t) + D_\psi(q_t^\circ, \overline{q}_t) - D_\psi(\tilde{q}_{t+1}, \overline{q}_t) - D_\psi(q_t^\circ, \tilde{q}_{t+1})$$

$$= D_\psi(\overline{q}_t, \tilde{q}_{t+1}) + D_\psi(q_t^\circ, \overline{q}_t) - D_\psi(q_t^\circ, \tilde{q}_{t+1})$$

Since only the optimality condition for the unconstrained problem and the three-point lemma have been employed so far, this result holds for any $q_t^\circ, \overline{q}_t$. Now, considering the result of the projection step of OMD $q_{t+1}$, the generalized Pythagorean Theorem allow us to state that $D_\psi(q_t^\circ, \tilde{q}_{t+1}) \geqslant D_\psi(q_t^\circ, q_{t+1})$. Such a result, together with the application of Lemma B.7 to $D_\psi(q_t^\circ, q_{t+1})$ provide:

$$\eta\langle \overline{q}_t - q_t^\circ; \ell_t\rangle = D_\psi(\overline{q}_t, \tilde{q}_{t+1}) + D_\psi(q_t^\circ, \overline{q}_t) - D_\psi(q_t^\circ, \tilde{q}_{t+1})$$

$$\leqslant D_\psi(\overline{q}_t, \tilde{q}_{t+1}) + D_\psi(q_t^\circ, \overline{q}_t) - D_\psi(q_t^\circ, q_{t+1})$$

$$\leqslant D_\psi(\overline{q}_t, \tilde{q}_{t+1}) + D_\psi(q_t^\circ, \overline{q}_t) - D_\psi(q_t^\circ, \overline{q}_{t+1}) + \frac{\alpha}{(1-\alpha)}D_\psi(q_t^\circ, u)$$

Now, summing over $t \in [\![T]\!]$ we get

$$\eta\overline{\mathcal{R}}_T^D = \eta \sum_{t=1}^T \langle \overline{q}_t - q_t^\circ; \ell_t\rangle \leqslant \underbrace{\sum_{t=1}^T D_\psi(\overline{q}_t, \tilde{q}_{t+1})}_{\substack{\text{``Stability''term} \\ \text{Lemma B.8}}} + \underbrace{\frac{\alpha}{(1-\alpha)} \sum_{t=1}^T D_\psi(q_t^\circ, u)}_{\substack{\text{``Residual'' term} \\ \text{Lemma B.9}}}$$

$$+ \underbrace{\sum_{t=1}^{T} \left( D_\psi(q_t^\circ, \overline{q}_t) - D_\psi(q_t^\circ, \overline{q}_{t+1}) \right)}_{\substack{\text{``Penalty'' term} \\ \text{Lemma C.5}}}.$$

Now we combine the results from Lemma B.8 for the "Stability" term, Lemma B.9 for the "Residual" term and Lemma C.5 for the "Penalty" term, obtaining:

$$\eta \overline{\mathcal{R}}_T^D \leqslant \eta^2 TL + L \log\left(X^2 A\right) + \mathcal{H}_T(\overline{D}_f^{\mathsf{P}}) + L^2 \log(2X^2 AT)\overline{D}_f^{\mathsf{P}} + 9L^2 \log\left(2X^2 AT\right)\log(T)$$

$$\square$$

## C.2 FURTHER LEMMAS

**Lemma C.5** ("Penalty" Bound). *For* $\overline{q}_t = (1-\alpha)q_t + \alpha u$, $\alpha = \frac{1}{T+1}$, $\epsilon_{t,k} = d_{TV}(q_t^{\circ,k}, q_{t-1}^{\circ,k})$, $\mathcal{H}_T(\overline{D}_f^{\mathsf{P}}) = TLh\left(\frac{(L^2+L)\overline{D}_f^{\mathsf{P}}}{2TL}\right)$ *and* $\overline{D}_f^{\mathsf{P}} = \sum_{t=2}^{T} \max_{(x,a)\in\mathcal{X}\times\mathcal{A}} \|\overline{P}_t(\cdot|x,a) - \overline{P}_{t-1}(\cdot|x,a)\|_1$ *it holds that*

$$\sum_{t=1}^{T} \left( D_\psi(q_t^\circ, \overline{q}_t) - D_\psi(q_t^\circ, \overline{q}_{t+1}) \right) \leqslant \mathcal{H}_T(\overline{D}_f^{\mathsf{P}}) + L^2 \log(2X^2 AT)\overline{D}_f^{\mathsf{P}} + 9L^2 \log\left(2X^2 AT\right)\log(T)$$

*Proof.* First, we unravel the summation:

$$\sum_{t=1}^{T} \left( D_\psi(q_t^\circ, \overline{q}_t) - D_\psi(q_t^\circ, \overline{q}_{t+1}) \right) = D_\psi(q_1^\circ, \overline{q}_1) - D_\psi(q_1^\circ, \overline{q}_2) + D_\psi(q_T^\circ, \overline{q}_T) - D_\psi(q_T^\circ, \overline{q}_{T+1})$$

$$\leqslant \underbrace{D_\psi(q_1^\circ, \overline{q}_1)}_{\substack{\text{Range of } \psi \\ \text{Lemma B.11}}} + \underbrace{\sum_{t=2}^{T} D_\psi(q_t^\circ, \overline{q}_t) - D_\psi(q_{t-1}^\circ, \overline{q}_t)}_{\substack{\text{non-telescoping term} \\ \text{Lemma C.6}}}$$

Where the inequality comes from neglecting negative terms. What is left to do is to combine Lemma B.11, which expresses the range of the negative Shannon Entropy, and Lemma C.6 providing a bound for the non-telescoping term.

$$\sum_{t=1}^{T} \left( D_\psi(q_t^\circ, \overline{q}_t) - D_\psi(q_t^\circ, \overline{q}_{t+1}) \right) \leqslant L \log\left(X^2 A\right) + \mathcal{H}_T(\overline{D}_f^{\mathsf{P}}) + L^2 \log(2X^2 AT)\overline{D}_f^{\mathsf{P}}$$

$$+ 8L^2 \log\left(2X^2 AT\right)\log(T)$$

$$\square$$

**Lemma C.6** (Bound on the non-telescoping term). *For* $q_t^\circ \in \Delta$, $\overline{q}_t = (1-\alpha)q_t + \alpha u$, $\epsilon_{t,k} = d_{TV}(q_t^{\circ,k}, q_{t-1}^{\circ,k})$, $\mathcal{H}_T(\overline{D}_f^{\mathsf{P}}) = TLh\left(\frac{(L^2+L)\overline{D}_f^{\mathsf{P}}}{2TL}\right)$ *and* $\overline{D}_f^{\mathsf{P}} = \sum_{t=2}^{T} \max_{(x,a)\in\mathcal{X}\times\mathcal{A}} \|\overline{P}_t(\cdot|x,a) - \overline{P}_{t-1}(\cdot|x,a)\|_1$ *we have that:*

$$\sum_{t=2}^{T} D_\psi(q_t^\circ, \overline{q}_t) - D_\psi(q_{t-1}^\circ, \overline{q}_t) \leqslant \mathcal{H}_T(\overline{D}_f^{\mathsf{P}}) + L^2 \log(2X^2 AT)\overline{D}_f^{\mathsf{P}} + 8L^2 \log\left(2X^2 AT\right)\log(T)$$

*Proof.* We first focus on each time-step $t$. From the definition of the Bregman Divergence we have:

$$D_\psi(q_t^\circ, \overline{q}_t) - D_\psi(q_{t-1}^\circ, \overline{q}_t) = \psi(q_t^\circ) - \psi(q_{t-1}^\circ) + \langle \nabla\psi(\overline{q}_t); q_{t-1}^\circ - q_t^\circ \rangle$$

$$= \psi(q_t^\circ) - \psi(q_{t-1}^\circ) + \langle \log(\overline{q}_t); q_{t-1}^\circ - q_t^\circ \rangle + \langle \mathbf{1}; q_{t-1}^\circ - q_t^\circ \rangle$$

$$\leqslant |\psi(q_t^\circ) - \psi(q_{t-1}^\circ)| + \|\log(\overline{q}_t)\|_\infty \|q_t^\circ - q_{t-1}^\circ\|_1$$

Where the second equality comes from the computation of the derivative of the Negative Shannon Entropy:

$$\nabla\psi(f) = \left(\frac{\partial\psi(f)}{\partial f}\right)^T = (\log(f) + 1, \ldots)^T$$

Finally, the last step comes from Holder's inequality and from taking the absolute value of the difference of the entropies.

We combine the results from Lemma C.7 and Lemma B.14 to obtain over the whole time horizon:

$$\sum_{t=2}^{T} D_\psi(q_t^\circ, \overline{q}_t) - D_\psi(q_{t-1}^\circ, \overline{q}_t) \leqslant \underbrace{\sum_{t=2}^{T} |\psi(q_t^\circ) - \psi(q_{t-1}^\circ)|}_{\text{Lemma C.7}} + \underbrace{\sum_{t=2}^{T} \|\log(\overline{q}_t)\|_\infty \|q_t^\circ - q_{t-1}^\circ\|_1}_{\text{Lemma B.14}}$$

$$\leqslant \mathcal{H}_T(\overline{D}_f^{\mathsf{P}}) + L^2 \log(2X^2 AT)\overline{D}_f^{\mathsf{P}}$$

$$+ 8L^2 \log\left(X^2 AT + X^2 A\right) \log(T)$$

$\square$

**Lemma C.7.** *For* $\mathcal{H}_T(\overline{D}_f^{\mathsf{P}}) = TLh\left(\frac{(L^2+L)\overline{D}_f^{\mathsf{P}}}{2TL}\right)$ *and* $\overline{D}_f^{\mathsf{P}} = \sum_{t=2}^{T} \max_{(x,a)\in\mathcal{X}\times\mathcal{A}} \|\overline{P}_t(\cdot|x,a) - \overline{P}_{t-1}(\cdot|x,a)\|_1$ *we have that:*

$$\sum_{t=2}^{T} \left|\psi\left(q_t^\circ\right) - \psi\left(q_{t-1}^\circ\right)\right| \leqslant \mathcal{H}_T(\overline{D}_f^{\mathsf{P}}) + L^2 \log(2X^2 AT)\overline{D}_f^{\mathsf{P}}.$$

*Proof.* Once defined $\epsilon_{t,k} = d_{\mathrm{TV}}(q_t^{\circ,k}, q_{t-1}^{\circ,k})$, $\mathcal{H}_T(\overline{D}_f^{\mathsf{P}}) = TLh\left(\frac{(L^2+L)\overline{D}_f^{\mathsf{P}}}{2TL}\right)$ and $\overline{D}_f^{\mathsf{P}} = \sum_{t=2}^{T} \max_{(x,a)\in\mathcal{X}\times\mathcal{A}} \|\overline{P}_t(\cdot|x,a) - \overline{P}_{t-1}(\cdot|x,a)\|_1$ the proof comes from a direct application of (Theorem 3, Sason, 2013):

$$\sum_{t=2}^{T} \left|\psi\left(q_t^\circ\right) - \psi\left(q_{t-1}^\circ\right)\right|$$

$$\leqslant \sum_{t=2}^{T} \sum_{k=0}^{L-1} \left|\psi\left(q_t^{\circ,k}\right) - \psi\left(q_{t-1}^{\circ,k}\right)\right|$$

$$\leqslant \sum_{t=2}^{T} \sum_{k=0}^{L-1} h\left(\epsilon_{t,k}\right) + \epsilon_{t,k} \log\left(X^2 A - 1\right)$$

$$\leqslant TL \sum_{t=2}^{T} \sum_{k=0}^{L-1} \frac{1}{TL} h\left(\epsilon_{t,k}\right) + \log\left(X^2 AT + X^2 A\right) \frac{1}{2} \sum_{t=2}^{T} \sum_{k=0}^{L-1} \|q_t^{\circ,k} - q_{t-1}^{\circ,k}\|_1$$

$$\leqslant TLh\left(\sum_{t=2}^{T} \sum_{k=0}^{L-1} \frac{1}{TL}\epsilon_{t,k}\right)$$

$$+ \frac{1}{2}(L^2 + L) \log\left(X^2 AT + X^2 A\right) \sum_{t=2}^{T} \max_{(x,a)\in\mathcal{X}\times\mathcal{A}} \|\overline{P}_t(\cdot|x,a) - \overline{P}_{t-1}(\cdot|x,a)\|_1$$

$$\leqslant TLh\left(\sum_{t=2}^{T} \sum_{k=0}^{L-1} \frac{\|q_t^{\circ,k} - q_{t-1}^{\circ,k}\|_1}{2TL}\right) + \frac{1}{2}(L^2 + L) \log\left(X^2 AT + X^2 A\right) \overline{D}_f^{\mathsf{P}}$$

$$\leqslant TLh\left(\sum_{t=2}^{T} \sum_{k=0}^{L-1} \frac{(L^2 + L) \sum_{t=2}^{T} \max_{(x,a)\in\mathcal{X}\times\mathcal{A}} \|\overline{P}_t(\cdot|x,a) - \overline{P}_{t-1}(\cdot|x,a)\|_1}{2TL}\right)$$

$$\leqslant TLh\left(\frac{(L^2+L)\overline{D}_f^{\mathsf{P}}}{2TL}\right) + \frac{1}{2}(L^2+L)\log\left(X^2AT + X^2A\right)\overline{D}_f^{\mathsf{P}}$$

$$\leqslant \mathcal{H}_T(\overline{D}_f^{\mathsf{P}}) + L^2\log(2X^2AT)\overline{D}_f^{\mathsf{P}}.$$

where the fourth inequality follows from Jensens' over the binary entropy. Assuming that $\frac{(L^2+L)\overline{D}_f^{\mathsf{P}}}{2TL} \leqslant 1/2$ and by applying Corollary F.10 we get the fifth inequality. Finally, we use the definition of $\mathcal{H}_T(\overline{D}_f^{\mathsf{P}})$ and further simplify the bound to achieve the final result. □

# D   THEORETICAL ANALYSIS WITH BANDIT-FEEDBACK AND AVERAGE SMOOTHING FUNCTIONS

In this section, we present the proofs of the results discussed in Section 5 with the specific smoothing functions $f_t(P_1, \ldots, P_t) = \frac{1}{t}\sum_{t'=1}^t P_t$. For convenience, we will restate the Theorems and Lemmas before providing a detailed analysis of each and report just the Lemmas that differentiate from Appendix B.

## D.1   MAIN RESULTS

**Corollary 5.2** (Smoothed-Regret Bound for SOMD under bandit-feedback and average smoothing). *Let $\eta = \sqrt{(21L\log(2X^2AT)(\log(T)))/(2XAT)}$, $\alpha = 1/(T+1)$, $\gamma = \eta$, smoothing functions such that $\overline{P}_t := \frac{1}{t}\sum_{t'\in\llbracket t\rrbracket} P_{t'}$ and any comparator policy $\pi^\circ \in \Pi$, Algorithm 2 suffers a smoothed regret of:*

$$\overline{\mathcal{R}}_T(\pi^\circ) \leqslant \mathcal{O}\left(L\overline{C}_f^{\mathsf{P}} + L^2\log(T) + L^{3/2}\sqrt{XAT\log\left(X^2AT\right)\log\left(T\right)}\right) \tag{17}$$

*Proof.* We first define $q_t = q_t^{\overline{P}_{t-1},\pi_t}$ and $q_t^\circ = q^{\overline{P}_t,\pi^\circ}$. It follows that the smoothed regret term can be decomposed as

$$\overline{\mathcal{R}}_T(\pi^\circ) \leqslant \sum_{t=1}^T \langle q^{\overline{P}_t,\pi_t} - q^{\overline{P}_t,\pi^\circ}; \ell_t\rangle = \underbrace{\mathbb{E}\left[\sum_{t=1}^T \langle q_t - q_t^\circ; \ell_t\rangle\right]}_{\substack{\text{Algorithmic Regret } (\overline{\mathcal{R}}_T^A) \\ \text{Lemma D.1}}} + \underbrace{\sum_{t=1}^T \langle q^{\overline{P}_t,\pi_t} - q^{\overline{P}_{t-1},\pi_t}; \ell_t\rangle}_{\text{Update Regret } (\overline{\mathcal{R}}_T^U)},$$

Where the expectation is with respect to the internal randomisation of the agent. Then, the result follows directly from the combination of Lemma B.1 for the Update Regret and Lemma D.1 for the Algorithmic Regret, namely:

$$\overline{\mathcal{R}}_T(\pi^\circ) \leqslant 2(L^2+L)(1+\log(T)) + 2L\overline{C}_f^{\mathsf{P}} + 4(L^2+L)(1+\log(T))$$

$$+ 14L\sqrt{LXAT\log\left(2X^2AT\right)\log(T)}$$

$$\leqslant 2L\overline{C}_f^{\mathsf{P}} + 12L^2 + 12L^2\log(T) + 14L\sqrt{LXAT\log\left(2X^2AT\right)\log(T)}$$

which leads to the final result. □

**Corollary 5.3** (Regret Bound for SOMD under bandit-feedback and average smoothing). *Let $\eta = \sqrt{(21L\log(2X^2AT)(\log(T)))/(2XAT)}$, $\alpha = 1/(T+1)$, $\gamma = \eta$, smoothing functions such that $\overline{P}_t := \frac{1}{t}\sum_{t'\in\llbracket t\rrbracket} P_{t'}$ and any comparator policy $\pi^\circ$, Algorithm 2 suffers a regret of:*

$$\mathcal{R}_T(\pi^\circ) = \tilde{\mathcal{O}}\left(L\overline{C}_f^{\mathsf{P}} + L^{3/2}\sqrt{XAT}\right). \tag{18}$$

*Proof.* The result follows straightforwardly from combining the results from Theorem 5.2 and Lemma 3.1. □

**Lemma D.1** (Algorithmic Regret Bound). *Choosing* $\eta = \sqrt{\frac{21L \log(2X^2 AT)(\log(T))}{2XAT}}$, $\alpha = \frac{1}{T+1}$ *and* $\gamma = \eta$ *the Algorithmic Regret is bounded by*

$$\overline{\mathcal{R}}_T^A = \mathbb{E}\left[\sum_{t=1}^{T}\langle q_t - q_t^\circ; \ell_t\rangle\right] \leqslant 2L\overline{C}_f^{\mathsf{P}} + 4(L^2 + L)(\log(T) + 1) + 14L\sqrt{LXAT \log(2X^2 AT)\log(T)}$$

*Proof.* The proof relies on the following decomposition:

$$\overline{\mathcal{R}}_T^A = \underbrace{\mathbb{E}\left[\sum_{t=1}^{T}\langle \overline{q}_t - q_t^\circ; \hat{\ell}_t\rangle\right]}_{\text{Descent Regret } (\overline{\mathcal{R}}_T^D)} + \underbrace{\mathbb{E}\left[\sum_{t=1}^{T}\langle q_t - \overline{q}_t; \hat{\ell}_t\rangle\right]}_{\text{Regularization Regret } (\overline{\mathcal{R}}_T^R)} + \underbrace{\mathbb{E}\left[\sum_{t=1}^{T}\langle q_t; \ell_t - \hat{\ell}_t\rangle\right]}_{\text{Bias 1 (B1)}} + \underbrace{\mathbb{E}\left[\sum_{t=1}^{T}\langle q_t^\circ; \hat{\ell}_t - \ell_t\rangle\right]}_{\text{Bias 2 (B2)}}$$

By applying Lemma D.2 for the Regularization Regret, Lemma D.3 for the Descent Regret and Lemmas D.5, D.6 for the bias terms respectively, we can rewrite the Algorithmic Regret as

$$\overline{\mathcal{R}}_T^A \leqslant \frac{L}{\eta} + \eta LXAT + 2(L^2 + L)(\log(T) + 1) + L\overline{C}_f^{\mathsf{P}} + \frac{1}{\eta}18L^2 \log(2X^2 AT)\log(T)$$

$$+ \frac{2L}{\eta} + \eta LXAT + L\overline{C}_f^{\mathsf{P}} + 2(L^2 + L)(\log(T) + 1)$$

$$\leqslant 2L\overline{C}_f^{\mathsf{P}} + 4(L^2 + L)(\log(T) + 1) + \eta 2LXAT + \frac{1}{\eta}21L^2 \log(2X^2 AT)\log(T)$$

We first chose the optimal $\eta$ as:

$$\eta = \sqrt{\frac{21L \log(2X^2 AT)(\log(T))}{2XAT}}$$

and then we upperbound the Algorithmic Regret after substituting the optimal $\eta$, namely:

$$\overline{\mathcal{R}}_T^A \leqslant 2L\overline{C}_f^{\mathsf{P}} + 4(L^2 + L)(\log(T) + 1) + \eta 2LXAT + \frac{1}{\eta}21L^2 \log(2X^2 AT)\log(T)$$

$$\leqslant 2L\overline{C}_f^{\mathsf{P}} + 4(L^2 + L)(\log(T) + 1) + 2L\sqrt{42LXAT \log(2X^2 AT)\log(T)}$$

$$\leqslant 2L\overline{C}_f^{\mathsf{P}} + 4(L^2 + L)(\log(T) + 1) + 14L\sqrt{LXAT \log(2X^2 AT)\log(T)}$$

which leads to the final result. $\square$

**Lemma D.2** (Regularisation Regret Bound). *For* $\overline{q}_t = (1 - \alpha)q_t + \alpha u$, $u(x, a, x') = \frac{1}{X_k \cdot A \cdot X_{k+1}}$ $\forall k \in [\![L-1]\!]$, $\alpha = \frac{1}{T+1}$ *and* $\gamma = \eta$, *it holds that*

$$\overline{\mathcal{R}}_T^R = \mathbb{E}\left[\sum_{t=1}^{T}\langle q_t - \overline{q}_t; \ell_t\rangle\right] \leqslant 2\frac{L}{\gamma} = 2\frac{L}{\eta}$$

*Proof.*

$$\overline{\mathcal{R}}_T^R = \sum_{t=1}^{T}\langle q_t - \overline{q}_t; \hat{\ell}_t\rangle = \alpha \sum_{t=1}^{T}\langle q_t - u; \hat{\ell}_t\rangle$$

$$\leqslant \frac{\alpha}{\gamma}\sum_{t=1}^{T}\|q_t - u\|_1$$

$$\leqslant \frac{\alpha}{\gamma}\sum_{t=1}^{T}\sum_{k=0}^{L-1}\|q_t^k - u\|_1$$

$$\leqslant 2\frac{\alpha}{\gamma}LT = \frac{1}{\gamma}\frac{2}{1+T}LT \leqslant 2\frac{L}{\gamma}$$

Where the first equality comes from the definition of $\overline{q}_t$, the first inequality follows from Holder's and the fact that $\|\hat{\ell}_t\|_\infty \leqslant \frac{1}{\gamma}$. The last from the trivial bound on the 1-norm between distributions and the definition of $\alpha = \frac{1}{T+1}$. $\square$

**Lemma D.3** (Descent Regret Bound). *For $\alpha = \frac{1}{T+1}$ the following fact holds*

$$\eta\overline{\mathcal{R}}_T^D = \eta\mathbb{E}\left[\sum_{t=1}^{T}\langle\overline{q}_t - q_t^{\circ}; \hat{\ell}_t\rangle\right] \leqslant \frac{\eta^2}{\gamma^2}L + \eta^2 LAXT + \frac{\eta^2}{\gamma}2(L^2 + L)(\log(T) + 1) + \frac{\eta^2}{\gamma}L\overline{C}_f^{\mathsf{P}}$$
$$+ 18L^2\log\left(2X^2AT\right)\log(T)$$

*Proof.* The proof follows the same steps as those in Lemma B.4 replacing $\ell_t$ with $\hat{\ell}_t$ to obtain,

$$\eta\overline{\mathcal{R}}_T^D = \eta\mathbb{E}\left[\sum_{t=1}^{T}\langle\overline{q}_t - q_t^{\circ}; \hat{\ell}_t\rangle\right]$$

$$\leqslant \mathbb{E}\left[\underbrace{\sum_{t=1}^{T}D_{\psi}(\overline{q}_t, \tilde{q}_{t+1})}_{\text{``Stability'' term}} + \underbrace{\frac{\alpha}{(1-\alpha)}\sum_{t=1}^{T}D_{\psi}(q_t^{\circ}, u)}_{\text{``Residual'' term}} + \underbrace{\sum_{t=1}^{T}(D_{\psi}(q_t^{\circ}, \overline{q}_t) - D_{\psi}(q_t^{\circ}, \overline{q}_{t+1}))}_{\text{``Penalty'' term}}\right].$$

Now we combine the results from Lemma D.4 for the "Stability" term, Lemma B.9 for the "Residual" term and Lemma B.10 for the "Penalty" term, obtaining:

$$\eta\overline{\mathcal{R}}_T^D \leqslant \frac{\eta^2}{\gamma^2}L + \eta^2 LAXT + \frac{\eta^2}{\gamma}2(L^2 + L)(\log(T) + 1) + \frac{\eta^2}{\gamma}L\overline{C}_f^{\mathsf{P}} + L\log\left(X^2A\right)$$
$$+ 17L^2\log\left(2X^2AT\right)\log(T)$$

$\square$

## D.2 AUXILIARY LEMMAS FOR THE BANDIT FEEDBACK

**Lemma D.4** ("Stability" Term Bound). *Choosing $\alpha = \frac{1}{T+1}$ and $\gamma > 0$,*

$$\mathbb{E}\left[\sum_{t=1}^{T}D_{\psi}(\overline{q}_t, \tilde{q}_{t+1})\right] \leqslant \frac{\eta^2}{\gamma^2}L + \eta^2 LAXT + \frac{\eta^2}{\gamma}2(L^2 + L)(\log(T) + 1) + \frac{\eta^2}{\gamma}L\overline{C}_f^{\mathsf{P}}$$

*Proof.* The term can be bounded as follows:

$$\mathbb{E}\left[\sum_{t=1}^{T}D_{\psi}(\overline{q}_t, \tilde{q}_{t+1})\right] = \mathbb{E}\left[\sum_{t=1}^{T}\left[\sum_{x,a,x'}\overline{q}_t(x,a,x')\log\left(\frac{\overline{q}_t(x,a,x')}{\tilde{q}_{t+1}(x,a,x')}\right) - \sum_{x,a,x'}\left(\overline{q}_t(x,a,x') - \tilde{q}_{t+1}(x,a,x')\right)\right]\right]$$

$$= \mathbb{E}\left[\sum_{t=1}^{T}\left[\sum_{x,a,x'}\eta\hat{\ell}_t(x,a)\overline{q}_t(x,a,x') + \overline{q}_t(x,a,x')\exp\left(-\eta\hat{\ell}_t(x,a)\right) - \overline{q}_t(x,a,x')\right]\right]$$

$$\leqslant \mathbb{E}\left[\sum_{t=1}^{T}\left[\sum_{x,a,x'}\eta^2\overline{q}_t(x,a,x')\mathbb{E}_t\left[\hat{\ell}_t^2(x,a)\right]\right]\right]$$

$$\leqslant (1-\alpha)\eta^2\sum_{t=1}^{T}\sum_{x,a}q^{\overline{P}_{t-1},\pi_t}(x,a)\frac{q^{P_t,\pi_t}(x,a)}{q^{P_t,\pi_t}(x,a)(q^{P_t,\pi_t}(x,a)+\gamma)}$$

$$+ \alpha\eta^2\sum_{t=1}^{T}\sum_{x,a}\frac{u(x,a)q^{P_t,\pi_t}(x,a)}{(q^{P_t,\pi_t}(x,a)+\gamma)^2}$$

$$\leqslant \frac{\eta^2}{\gamma^2}\alpha TL + \eta^2\sum_{t=1}^{T}\sum_{x,a}\frac{q^{\overline{P}_{t-1},\pi_t}(x,a)}{(q^{P_t,\pi_t}(x,a)+\gamma)}$$

$$\leqslant \frac{\eta^2}{\gamma^2}L + \eta^2\sum_{t=1}^{T}\sum_{x,a}\frac{\left|q^{\overline{P}_{t-1},\pi_t}(x,a) \pm q^{\overline{P}_t,\pi_t}(x,a) \pm q^{P_t,\pi_t}(x,a)\right|}{(q^{P_t,\pi_t}(x,a)+\gamma)}$$

$$\leqslant \frac{\eta^2}{\gamma^2}L + \eta^2 \sum_{t=1}^{T}\sum_{x,a} \frac{\left|q^{\overline{P}_{t-1},\pi_t}(x,a) - q^{\overline{P}_t,\pi_t}(x,a)\right|}{(q^{P_t,\pi_t}(x,a) + \gamma)}$$

$$+ \eta^2 \sum_{t=1}^{T}\sum_{x,a} \frac{\left|q^{\overline{P}_t,\pi_t}(x,a) - q^{P_t,\pi_t}(x,a)\right|}{(q^{P_t,\pi_t}(x,a) + \gamma)} + \eta^2 \sum_{t=1}^{T}\sum_{x,a} \frac{q^{P_t,\pi_t}(x,a)}{(q^{P_t,\pi_t}(x,a) + \gamma)}$$

$$\leqslant \frac{\eta^2}{\gamma^2}L + \eta^2 LXAT + \frac{\eta^2}{\gamma} \sum_{t=1}^{T}\sum_{x,a} \left|q^{\overline{P}_t,\pi_t}(x,a) - q^{P_t,\pi_t}(x,a)\right|$$

$$+ \frac{\eta^2}{\gamma} \sum_{t=1}^{T}\sum_{x,a} \left|q^{\overline{P}_{t-1},\pi_t}(x,a) - q^{\overline{P}_t,\pi_t}(x,a)\right|$$

$$\leqslant \frac{\eta^2}{\gamma^2}L + \eta^2 LXAT + \frac{\eta^2}{\gamma} \sum_{t=1}^{T}\sum_{x} \left|q^{\overline{P}_t,\pi_t}(x) - q^{P_t,\pi_t}(x)\right|$$

$$+ \frac{\eta^2}{\gamma} \sum_{t=1}^{T} \left\|q^{\overline{P}_{t-1},\pi_t} - q^{\overline{P}_t,\pi_t}\right\|_1$$

$$\leqslant \frac{\eta^2}{\gamma^2}L + \eta^2 LXAT + \frac{\eta^2}{\gamma} L\overline{C}_f^{\mathsf{P}} + \frac{\eta^2}{\gamma} 2(L^2 + L)(\log(T) + 1)$$

Where the first equality comes from the definition of generalized KL divergence, the second by applying the definition of the solution of the unconstrained optimization problem, namely:

$$\tilde{q}_{t+1}(x,a,x') = \overline{q}_t(x,a,x') \exp\left(-\eta\hat{\ell}_t(x,a)\right), \forall (x,a,x') \in \mathcal{X} \times \mathcal{A} \times \mathcal{X}.$$

The first inequality comes from the standard bound of the exponential function,

$$e^{-\eta\hat{\ell}_t(x,a)} \leqslant 1 - \eta\hat{\ell}_t(x,a) + \left(\eta\hat{\ell}_t(x,a)\right)^2, \forall\, \eta\hat{\ell}_t(x,a) \geqslant 0$$

which is satisfied $\forall \gamma > 0$. The fifth inequality comes from setting $\alpha = \frac{1}{T+1}$ and from triangle inequality. Finally, the last step comes from Corollary F.6 and Lemma B.1. $\qquad\square$

**Lemma D.5** (Bias 1 Bound). *For $\hat{\ell}_t$ as in Algorithm 2 it holds that,*

$$(BIAS\ 1)\quad \mathbb{E}\left[\sum_{t=1}^{T}\langle q^{\overline{P}_{t-1},\pi_t}; \ell_t - \hat{\ell}_t\rangle\right] \leqslant \gamma LXAT + L\overline{C}_f^{\mathsf{P}} + 2(L^2 + L)(\log(T) + 1)$$

*Proof.*

$$\mathbb{E}\left[\sum_{t=1}^{T}\langle q^{\overline{P}_{t-1},\pi_t}; \ell_t - \hat{\ell}_t\rangle\right] = \mathbb{E}\left[\sum_{t=1}^{T}\sum_{x,a} q^{\overline{P}_{t-1},\pi_t}(x,a)(\ell_t(x,a) - \mathbb{E}_t\left[\hat{\ell}_t(x,a)\right])\right]$$

$$= \sum_{t=1}^{T}\sum_{x,a} q^{\overline{P}_{t-1},\pi_t}(x,a)\ell_t(x,a)\left(\frac{\gamma}{q^{P_t,\pi_t}(x,a) + \gamma}\right)$$

$$\pm \sum_{t=1}^{T}\sum_{x,a} q^{P_t,\pi_t}(x,a)\ell_t(x,a)\left(\frac{\gamma}{q^{P_t,\pi_t}(x,a) + \gamma}\right)$$

$$\leqslant \sum_{t=1}^{T}\sum_{x,a} q^{P_t,\pi_t}(x,a)\ell_t(x,a)\left(\frac{\gamma}{q^{P_t,\pi_t}(x,a) + \gamma}\right)$$

$$+ \sum_{t=1}^{T}\sum_{x,a} \left(q^{\overline{P}_{t-1},\pi_t}(x,a) - q^{P_t,\pi_t}(x,a)\right)\ell_t(x,a)\left(\frac{\gamma}{q^{P_t,\pi_t}(x,a) + \gamma}\right)$$

$$\leqslant \sum_{t=1}^{T}\sum_{x,a} q^{P_t,\pi_t}(x,a)\ell_t(x,a)\left(\frac{\gamma}{q^{P_t,\pi_t}(x,a)+\gamma}\right)$$

$$+\sum_{t=1}^{T}\sum_{x,a}\left|q^{\overline{P}_{t-1},\pi_t}(x,a)-q^{P_t,\pi_t}(x,a)\right|\left(\frac{\gamma}{q^{P_t,\pi_t}(x,a)+\gamma}\right)$$

$$\leqslant \gamma LXAT + \sum_{t=1}^{T}\sum_{x,a}\left|q^{\overline{P}_{t-1},\pi_t}(x,a)-q^{P_t,\pi_t}(x,a)\right|$$

$$\leqslant \gamma LXAT + \sum_{t=1}^{T}\sum_{x,a}\left|q^{\overline{P}_{t-1},\pi_t}(x,a)-q^{P_t,\pi_t}(x,a)\pm q^{\overline{P}_t,\pi_t}(x,a)\right|$$

$$\leqslant \gamma LXAT + \sum_{t=1}^{T}\sum_{x,a}\left|q^{\overline{P}_{t-1},\pi_t}(x,a)-q^{\overline{P}_t,\pi_t}(x,a)\right|$$

$$+\sum_{t=1}^{T}\sum_{x,a}\left|q^{\overline{P}_t,\pi_t}(x,a)-q^{P_t,\pi_t}(x,a)\right|$$

$$\leqslant \gamma LXAT + \sum_{t=1}^{T}\left\|q^{\overline{P}_{t-1},\pi_t}-q^{\overline{P}_t,\pi_t}\right\|_1$$

$$+\sum_{t=1}^{T}\sum_{x}\left|q^{\overline{P}_t,\pi_t}(x)-q^{P_t,\pi_t}(x)\right|$$

$$\leqslant \gamma LXAT + 2(L^2+L)(\log(T)+1)+L\overline{C}_f^{\mathsf{P}}$$

Where the last inequality comes from applying Lemma B.1 and Corollary F.6. $\qquad\square$

**Lemma D.6** (Bias 2 Bound). *For $\hat{\ell}_t$ as in Algorithm 2 it holds that,*

$$(BIAS\ 2)\quad \mathbb{E}\left[\sum_{t=1}^{T}\langle q_t^{\overline{P}_t,\pi^\circ};\hat{\ell}_t-\ell_t\rangle\right]\leqslant 0$$

*Proof.* It is sufficient to recall that,

$$\ell_t(x,a)-\mathbb{E}_t\left[\hat{\ell}_t(x,a)\right]\in\left[0,\frac{\gamma\ell_t(x,a)}{q^{P_t,\pi_t(x,a)}}\right]$$

Namely, that we are underestimating the true loss. $\qquad\square$

# E  THEORETICAL ANALYSIS WITH BANDIT-FEEDBACK AND GENERIC SMOOTHING FUNCTION

In this section, we present the proofs of the results discussed in Section 5 related to the regret analysis agnostic of the smoothing function to be used. For convenience, we will restate the Theorems and Lemmas before providing a detailed analysis of each and report just the Lemmas that differentiate from Appendix B and Appendix D.

## E.1  MAIN RESULTS

**Theorem 5.1** (Smoothed-Regret Bound for SOMD under bandit-feedback). *Let $\eta = \sqrt{(13L\log(2X^2AT)\rho_T^f)/(2XAT)}$, $\alpha = 1/(T+1)$, $\gamma = \eta$, generic smoothing functions and any comparator policy $\pi^\circ \in \Pi$, Algorithm 2 suffers a smoothed regret of:*

$$\overline{\mathcal{R}}_T(\pi^\circ)\leqslant \mathcal{O}\left(L^2\overline{D}_f^{\mathsf{P}}+L\overline{C}_f^{\mathsf{P}}+L^{3/2}\sqrt{XAT\rho_T^f\log(X^2AT)}\right). \tag{16}$$

*Proof.* We first define $q_t = q_t^{\overline{P}_{t-1}, \pi_t}$ and $q_t^\circ = q^{\overline{P}_t, \pi^\circ}$. It follows that the smoothed regret term can be decomposed as

$$\overline{\mathcal{R}}_T(\pi^\circ) \leqslant \sum_{t=1}^T \langle q^{\overline{P}_t, \pi_t} - q^{\overline{P}_t, \pi^\circ}; \ell_t \rangle = \mathbb{E}\underbrace{\left[ \sum_{t=1}^T \langle q_t - q_t^\circ; \ell_t \rangle \right]}_{\substack{\text{Algorithmic Regret } (\overline{\mathcal{R}}_T^A) \\ \text{Lemma E.1}}} + \underbrace{\sum_{t=1}^T \langle q^{\overline{P}_t, \pi_t} - q^{\overline{P}_{t-1}, \pi_t}; \ell_t \rangle}_{\text{Update Regret } (\overline{\mathcal{R}}_T^U)},$$

Where the expectation is with respect to the internal randomisation of the agent. Then, the result follows directly from the combination of Lemma C.2 for the Update Regret and Lemma E.1 for the Algorithmic Regret, namely:

$$\overline{\mathcal{R}}_T(\pi^\circ) \leqslant (L^2 + L)\overline{D}_f^{\mathsf{P}} + 2(L^2 + L)\overline{D}_f^{\mathsf{P}} + 2L\overline{C}_f^{\mathsf{P}} + 12L\sqrt{LXAT \log(2X^2AT)\rho_T^f}$$

which leads to the final result. $\qquad\square$

**Lemma E.1** (Algorithmic Regret Bound). *Choosing* $\eta = \sqrt{\frac{13L \log(2X^2AT)\rho_T^f}{2XAT}}$, $\alpha = \frac{1}{T+1}$, $\gamma = \eta$,
$\mathcal{H}_T(\overline{D}_f^{\mathsf{P}}) = TLh\left( \frac{(L^2+L)\overline{D}_f^{\mathsf{P}}}{2TL} \right)$, $\overline{D}_f^{\mathsf{P}} = \sum_{t=2}^T \max_{(x,a) \in \mathcal{X} \times \mathcal{A}} \|\overline{P}_t(\cdot|x, a) - \overline{P}_{t-1}(\cdot|x, a)\|_1$ *and for*
$\rho_T^f = \log(T) + \overline{D}_f^{\mathsf{P}} + \mathcal{H}_f$ ,*the Algorithmic Regret is bounded by*

$$\overline{\mathcal{R}}_T^A = \mathbb{E}\left[ \sum_{t=1}^T \langle q_t - q_t^\circ; \ell_t \rangle \right] \leqslant 2(L^2 + L)\overline{D}_f^{\mathsf{P}} + 2L\overline{C}_f^{\mathsf{P}} + 12L\sqrt{LXAT \log(2X^2AT)\rho_T^f}$$

*Proof.* The proof relies on the following decomposition:

$$\overline{\mathcal{R}}_T^A = \underbrace{\mathbb{E}\left[ \sum_{t=1}^T \langle \overline{q}_t - q_t^\circ; \hat{\ell}_t \rangle \right]}_{\text{Descent Regret } (\overline{\mathcal{R}}_T^D)} + \underbrace{\mathbb{E}\left[ \sum_{t=1}^T \langle q_t - \overline{q}_t; \hat{\ell}_t \rangle \right]}_{\text{Regularization Regret } (\overline{\mathcal{R}}_T^R)} + \underbrace{\mathbb{E}\left[ \sum_{t=1}^T \langle q_t; \ell_t - \hat{\ell}_t \rangle \right]}_{\text{Bias 1 (B1)}} + \underbrace{\mathbb{E}\left[ \sum_{t=1}^T \langle q_t^\circ; \hat{\ell}_t - \ell_t \rangle \right]}_{\text{Bias 2 (B2)}}$$

By applying Lemma D.2 for the Regularization Regret, Lemma E.2 for the Descent Regret and Lemmas E.4, D.6 for the bias terms, we can rewrite the Algorithmic Regret as

$$\overline{\mathcal{R}}_T^A \leqslant 2\frac{L}{\eta} + \frac{\eta}{\gamma^2}L + \eta LXAT + \frac{\eta}{\gamma}L\overline{C}_f^{\mathsf{P}} + \frac{\eta}{\gamma}(L^2 + L)\overline{D}_f^{\mathsf{P}} + \gamma LXAT + L\overline{C}_f^{\mathsf{P}} + (L^2 + L)\overline{D}_f^{\mathsf{P}}$$

$$+ \frac{1}{\eta}\left( \mathcal{H}_T(\overline{D}_f^{\mathsf{P}}) + L^2 \log(2X^2AT)\overline{D}_f^{\mathsf{P}} + 10L^2 \log\left(2X^2AT\right) \log(T) \right)$$

$$\leqslant 2\frac{L}{\eta} + \frac{1}{\eta}L + \eta LXAT + L\overline{C}_f^{\mathsf{P}} + (L^2 + L)\overline{D}_f^{\mathsf{P}} + \eta LXAT + L\overline{C}_f^{\mathsf{P}} + (L^2 + L)\overline{D}_f^{\mathsf{P}}$$

$$+ \frac{1}{\eta}\left( \mathcal{H}_T(\overline{D}_f^{\mathsf{P}}) + L^2 \log(2X^2AT)\overline{D}_f^{\mathsf{P}} + 10L^2 \log\left(2X^2AT\right) \log(T) \right)$$

$$\leqslant \eta 2LXAT + 2L\overline{C}_f^{\mathsf{P}} + 2(L^2 + L)\overline{D}_f^{\mathsf{P}}$$

$$+ \frac{1}{\eta}\left( \mathcal{H}_T(\overline{D}_f^{\mathsf{P}}) + L^2 \log(2X^2AT)\overline{D}_f^{\mathsf{P}} + 13L^2 \log\left(2X^2AT\right) \log(T) \right)$$

$$\leqslant \eta 2LXAT + 2L\overline{C}_f^{\mathsf{P}} + 2(L^2 + L)\overline{D}_f^{\mathsf{P}}$$

$$+ \frac{1}{\eta}13L^2 \log\left(2X^2AT\right)\left( \mathcal{H}_T(\overline{D}_f^{\mathsf{P}}) + \overline{D}_f^{\mathsf{P}} + \log(T) \right)$$

$$\leqslant \eta 2LXAT + 2(L^2 + L)\overline{D}_f^{\mathsf{P}} + 2L\overline{C}_f^{\mathsf{P}} + \frac{13}{\eta}L^2 \log\left(2X^2AT\right)\rho_T^f$$

We first chose the optimal $\eta$ as:

$$\eta = \sqrt{\frac{13L \log(2X^2AT)\rho_T^f}{2XAT}}$$

and then we upperbound the Algorithmic Regret after substituting the optimal $\eta$, namely:

$$\overline{\mathcal{R}}_T^A \leqslant 2(L^2 + L)\overline{D}_f^{\mathsf{P}} + 2L\overline{C}_f^{\mathsf{P}} + 2L\sqrt{26LXAT\log(2X^2AT)\rho_T^f}$$

$$\leqslant 2(L^2 + L)\overline{D}_f^{\mathsf{P}} + 2L\overline{C}_f^{\mathsf{P}} + 12L\sqrt{LXAT\log(2X^2AT)\rho_T^f}$$

which leads to the final result. $\qquad\square$

**Lemma E.2** (Descent Regret Bound). *For $\alpha = \frac{1}{T+1}$ and defining $\mathcal{H}_T(\overline{D}_f^{\mathsf{P}}) = TLh\left(\frac{(L^2+L)\overline{D}_f^{\mathsf{P}}}{2TL}\right)$,
$\overline{D}_f^{\mathsf{P}} = \sum_{t=2}^T \max_{(x,a)\in\mathcal{X}\times\mathcal{A}} \|\overline{P}_t(\cdot|x,a) - \overline{P}_{t-1}(\cdot|x,a)\|_1$, it holds that:*

$$\eta\overline{\mathcal{R}}_T^D = \eta\mathbb{E}\left[\sum_{t=1}^T \langle \overline{q}_t - q_t^{\circ}; \hat{\ell}_t \rangle\right] \leqslant \frac{\eta^2}{\gamma^2}L + \eta^2 LXAT + \frac{\eta^2}{\gamma}L\overline{C}_f^{\mathsf{P}} + \frac{\eta^2}{\gamma}(L^2 + L)\overline{D}_f^{\mathsf{P}}$$

$$+ \mathcal{H}_T(\overline{D}_f^{\mathsf{P}}) + L^2\log(2X^2AT)\overline{D}_f^{\mathsf{P}} + 10L^2\log\left(2X^2AT\right)\log(T)$$

*Proof.* The proof follows the same steps as those in Lemma C.4 replacing $\ell_t$ with $\hat{\ell}_t$ to obtain,

$$\eta\overline{\mathcal{R}}_T^D = \eta\mathbb{E}\left[\sum_{t=1}^T \langle \overline{q}_t - q_t^{\circ}; \hat{\ell}_t \rangle\right]$$

$$\leqslant \mathbb{E}\left[\underbrace{\sum_{t=1}^T D_\psi(\overline{q}_t, \tilde{q}_{t+1})}_{\text{Stability term}} + \underbrace{\frac{\alpha}{(1-\alpha)}\sum_{t=1}^T D_\psi(q_t^{\circ}, u)}_{\text{Residual term}} + \underbrace{\sum_{t=1}^T (D_\psi(q_t^{\circ}, \overline{q}_t) - D_\psi(q_t^{\circ}, \overline{q}_{t+1}))}_{\text{Penalty term}}\right].$$

Now we combine the results from Lemma E.3 for the "Stability" term, Lemma B.9 for the "Residual" term and Lemma C.5 for the "Penalty" term, obtaining:

$$\eta\overline{\mathcal{R}}_T^D \leqslant \frac{\eta^2}{\gamma^2}L + \eta^2 LXAT + \frac{\eta^2}{\gamma}L\overline{C}_f^{\mathsf{P}} + \frac{\eta^2}{\gamma}(L^2 + L)\overline{D}_f^{\mathsf{P}}$$

$$+ \mathcal{H}_T(\overline{D}_f^{\mathsf{P}}) + L^2\log(2X^2AT)\overline{D}_f^{\mathsf{P}} + 10L^2\log\left(2X^2AT\right)\log(T)$$

$$\square$$

### E.2 Auxiliary Lemmas for the Bandit Feedback

**Lemma E.3** (Bound of "Stability" term). *Chosing $\alpha = \frac{1}{T+1}$, $\gamma > 0$ and defining $\mathcal{H}_T(\overline{D}_f^{\mathsf{P}}) = TLh\left(\frac{(L^2+L)\overline{D}_f^{\mathsf{P}}}{2TL}\right)$, $\overline{D}_f^{\mathsf{P}} = \sum_{t=2}^T \max_{(x,a)\in\mathcal{X}\times\mathcal{A}} \|\overline{P}_t(\cdot|x,a) - \overline{P}_{t-1}(\cdot|x,a)\|_1$, it holds that:*

$$\mathbb{E}\left[\sum_{t=1}^T D_\psi(\overline{q}_t, \tilde{q}_{t+1})\right] \leqslant \frac{\eta^2}{\gamma^2}L + \eta^2 LXAT + \frac{\eta^2}{\gamma}L\overline{C}_f^{\mathsf{P}} + \frac{\eta^2}{\gamma}(L^2 + L)\overline{D}_f^{\mathsf{P}}$$

*Proof.* The term can be bounded as follows:

$$\mathbb{E}\left[\sum_{t=1}^T D_\psi(\overline{q}_t, \tilde{q}_{t+1})\right] = \mathbb{E}\left[\sum_{t=1}^T \left[\sum_{x,a,x'} \overline{q}_t(x,a,x')\log\left(\frac{\overline{q}_t(x,a,x')}{\tilde{q}_{t+1}(x,a,x')}\right) - \sum_{x,a,x'} \left(\overline{q}_t(x,a,x') - \tilde{q}_{t+1}(x,a,x')\right)\right]\right]$$

$$= \mathbb{E}\left[\sum_{t=1}^T \left[\sum_{x,a,x'} \eta\hat{\ell}_t(x,a)\overline{q}_t(x,a,x') + \overline{q}_t(x,a,x')\exp\left(-\eta\hat{\ell}_t(x,a)\right) - \overline{q}_t(x,a,x')\right]\right]$$

$$\leqslant \mathbb{E}\left[\sum_{t=1}^T \left[\sum_{x,a,x'} \eta^2\overline{q}_t(x,a,x')\mathbb{E}_t\left[\hat{\ell}_t^2(x,a)\right]\right]\right]$$

$$\leqslant (1-\alpha)\eta^2 \sum_{t=1}^{T} \sum_{x,a} q^{\overline{P}_{t-1},\pi_t}(x,a) \frac{q^{P_t,\pi_t}(x,a)}{q^{P_t,\pi_t}(x,a)(q^{P_t,\pi_t}(x,a)+\gamma)}$$

$$+ \alpha\eta^2 \sum_{t=1}^{T} \sum_{x,a} \frac{u(x,a)q^{P_t,\pi_t}(x,a)}{(q^{P_t,\pi_t}(x,a)+\gamma)^2}$$

$$\leqslant \frac{\eta^2}{\gamma^2}\alpha T L + \eta^2 \sum_{t=1}^{T} \sum_{x,a} \frac{q^{\overline{P}_{t-1},\pi_t}(x,a)}{(q^{P_t,\pi_t}(x,a)+\gamma)}$$

$$\leqslant \frac{\eta^2}{\gamma^2} L + \eta^2 \sum_{t=1}^{T} \sum_{x,a} \frac{\left|q^{\overline{P}_{t-1},\pi_t}(x,a) \pm q^{\overline{P}_t,\pi_t}(x,a) \pm q^{P_t,\pi_t}(x,a)\right|}{(q^{P_t,\pi_t}(x,a)+\gamma)}$$

$$\leqslant \frac{\eta^2}{\gamma^2} L + \eta^2 \sum_{t=1}^{T} \sum_{x,a} \frac{\left|q^{\overline{P}_{t-1},\pi_t}(x,a) - q^{\overline{P}_t,\pi_t}(x,a)\right|}{(q^{P_t,\pi_t}(x,a)+\gamma)}$$

$$+ \eta^2 \sum_{t=1}^{T} \sum_{x,a} \frac{\left|q^{\overline{P}_t,\pi_t}(x,a) - q^{P_t,\pi_t}(x,a)\right|}{(q^{P_t,\pi_t}(x,a)+\gamma)} + \eta^2 \sum_{t=1}^{T} \sum_{x,a} \frac{q^{P_t,\pi_t}(x,a)}{(q^{P_t,\pi_t}(x,a)+\gamma)}$$

$$\leqslant \frac{\eta^2}{\gamma^2} L + \eta^2 L X A T + \frac{\eta^2}{\gamma} \sum_{t=1}^{T} \sum_{x,a} \left|q^{\overline{P}_t,\pi_t}(x,a) - q^{P_t,\pi_t}(x,a)\right|$$

$$+ \frac{\eta^2}{\gamma} \sum_{t=1}^{T} \sum_{x,a} \left|q^{\overline{P}_{t-1},\pi_t}(x,a) - q^{\overline{P}_t,\pi_t}(x,a)\right|$$

$$\leqslant \frac{\eta^2}{\gamma^2} L + \eta^2 L X A T + \frac{\eta^2}{\gamma} \sum_{t=1}^{T} \sum_{x} \left|q^{\overline{P}_t,\pi_t}(x) - q^{P_t,\pi_t}(x)\right|$$

$$+ \frac{\eta^2}{\gamma} \sum_{t=1}^{T} \left\|q^{\overline{P}_{t-1},\pi_t} - q^{\overline{P}_t,\pi_t}\right\|_1$$

$$\leqslant \frac{\eta^2}{\gamma^2} L + \eta^2 L X A T + \frac{\eta^2}{\gamma} L \overline{C}_f^{\mathsf{P}} + \frac{\eta^2}{\gamma}(L^2+L)\overline{D}_f^{\mathsf{P}}$$

Where the first equality comes from the definition of generalized KL divergence, the second by applying the definition of the solution of the unconstrained optimization problem, namely:

$$\tilde{q}_{t+1}(x,a,x') = \overline{q}_t(x,a,x')\exp\left(-\eta\hat{\ell}_t(x,a)\right), \forall (x,a,x') \in \mathcal{X} \times \mathcal{A} \times \mathcal{X}$$

and further simplifications. The first inequality comes from the standard bound of the exponential function,

$$e^{-\eta\hat{\ell}_t(x,a)} \leqslant 1 - \eta\hat{\ell}_t(x,a) + \left(\eta\hat{\ell}_t(x,a)\right)^2, \forall \, \eta\hat{\ell}_t(x,a) \geqslant 0$$

which is satisfied $\forall \gamma > 0$. The fourth inequality comes from setting $\alpha = \frac{1}{T+1}$ and from triangle inequality. Finally, the last inequality comes from Corollary F.6 and Lemma C.2. $\qquad\square$

**Lemma E.4** (Bound on Bias 1). *For $\gamma > 0$ and defining $\overline{D}_f^{\mathsf{P}} = \sum_{t=2}^{T} \max_{(x,a)\in\mathcal{X}\times\mathcal{A}} \|\overline{P}_t(\cdot|x,a) - \overline{P}_{t-1}(\cdot|x,a)\|_1$, it holds that:*

$$\mathbb{E}\left[\sum_{t=1}^{T} \langle q^{\overline{P}_{t-1},\pi_t}; \ell_t - \hat{\ell}_t\rangle\right] \leqslant \gamma L X A T + L\overline{C}_f^{\mathsf{P}} + (L^2+L)\overline{D}_f^{\mathsf{P}}$$

*Proof.*

$$\mathbb{E}\left[\sum_{t=1}^{T} \langle q^{\overline{P}_{t-1},\pi_t}; \ell_t - \hat{\ell}_t\rangle\right] = \mathbb{E}\left[\sum_{t=1}^{T} \sum_{x,a} q^{\overline{P}_{t-1},\pi_t}(x,a)(\ell_t(x,a) - \mathbb{E}_t\left[\hat{\ell}_t(x,a)\right])\right]$$

$$= \sum_{t=1}^{T} \sum_{x,a} q^{\overline{P}_{t-1},\pi_t}(x,a) \ell_t(x,a) \left( \frac{\gamma}{q^{P_t,\pi_t}(x,a) + \gamma} \right)$$

$$\pm \sum_{t=1}^{T} \sum_{x,a} q^{P_t,\pi_t}(x,a) \ell_t(x,a) \left( \frac{\gamma}{q^{P_t,\pi_t}(x,a) + \gamma} \right)$$

$$\leqslant \sum_{t=1}^{T} \sum_{x,a} q^{P_t,\pi_t}(x,a) \ell_t(x,a) \left( \frac{\gamma}{q^{P_t,\pi_t}(x,a) + \gamma} \right)$$

$$+ \sum_{t=1}^{T} \sum_{x,a} \left( q^{\overline{P}_{t-1},\pi_t}(x,a) - q^{P_t,\pi_t}(x,a) \right) \ell_t(x,a) \left( \frac{\gamma}{q^{P_t,\pi_t}(x,a) + \gamma} \right)$$

$$\leqslant \sum_{t=1}^{T} \sum_{x,a} q^{P_t,\pi_t}(x,a) \ell_t(x,a) \left( \frac{\gamma}{q^{P_t,\pi_t}(x,a) + \gamma} \right)$$

$$+ \sum_{t=1}^{T} \sum_{x,a} \left| q^{\overline{P}_{t-1},\pi_t}(x,a) - q^{P_t,\pi_t}(x,a) \right| \left( \frac{\gamma}{q^{P_t,\pi_t}(x,a) + \gamma} \right)$$

$$\leqslant \gamma L X A T + \sum_{t=1}^{T} \sum_{x,a} \left| q^{\overline{P}_{t-1},\pi_t}(x,a) - q^{P_t,\pi_t}(x,a) \right|$$

$$\leqslant \gamma L X A T + \sum_{t=1}^{T} \sum_{x,a} \left| q^{\overline{P}_{t-1},\pi_t}(x,a) - q^{P_t,\pi_t}(x,a) \pm q^{\overline{P}_t,\pi_t}(x,a) \right|$$

$$\leqslant \gamma L X A T + \sum_{t=1}^{T} \sum_{x,a} \left| q^{\overline{P}_{t-1},\pi_t}(x,a) - q^{\overline{P}_t,\pi_t}(x,a) \right|$$

$$+ \sum_{t=1}^{T} \sum_{x,a} \left| q^{\overline{P}_t,\pi_t}(x,a) - q^{P_t,\pi_t}(x,a) \right|$$

$$\leqslant \gamma L X A T + \sum_{t=1}^{T} \left\| q^{\overline{P}_{t-1},\pi_t} - q^{\overline{P}_t,\pi_t} \right\|_1 + \sum_{t=1}^{T} \sum_{x} \left| q^{\overline{P}_t,\pi_t}(x) - q^{P_t,\pi_t}(x) \right|$$

$$\leqslant \gamma L X A T + (L^2 + L) \overline{D}_f^{\mathsf{P}} + L \overline{C}_f^{\mathsf{P}}$$

Where the last inequality comes from applying Lemma C.2 and Corollary F.6. $\qquad\square$

# F INSTRUMENTAL LEMMAS

Here we report a few additional instrumental lemmas used throughout the proofs together with known lemmas from some references.

## F.1 STATISTICAL PROPERTIES OF $\hat{\ell}_t$

**Lemma F.1** (Bias of $\hat{\ell}_t$). *Given the estimator used by Algorithm 2, and for $\gamma > 0$ we have that,*

$$\ell_t(x,a) - \mathbb{E}_t \left[ \ell_t(x,a) \right] \leqslant \frac{\gamma \ell_t(x,a)}{q^{P_t,\pi_t}(x,a)}$$

*Proof.*

$$\ell_t(x,a) - \mathbb{E}_t \left[ \ell_t(x,a) \right] = \ell_t(x,a) - \frac{\ell_t(x,a) \mathbb{E}_t \left[ \mathbb{1}_t(x,a) \right]}{q^{P_t,\pi_t}(x,a) + \gamma}$$

$$= \ell_t(x,a) \left( 1 - \frac{q^{P_t,\pi_t}(x,a)}{q^{P_t,\pi_t}(x,a) + \gamma} \right)$$

$$= \ell_t(x,a) \left( 1 - \frac{q^{P_t, \pi_t}(x,a)}{q^{P_t, \pi_t}(x,a) + \gamma} \right)$$

$$\leqslant \frac{\gamma \ell_t(x,a)}{q^{P_t, \pi_t}(x,a)}$$

$\square$

**Lemma F.2** (Second-order moment of $\hat{\ell}_t$). *Given the estimator used by Algorithm 2, and for $\gamma > 0$ we have that,*

$$\mathbb{E}_t[\hat{\ell}_t^2(x,a)] \leqslant \frac{\ell_t^2(x,a)}{(q^{P_t, \pi_t}(x,a) + \gamma)}$$

*Proof.*

$$\mathbb{E}_t[\hat{\ell}_t^2(x,a)] \leqslant \left( \frac{\ell_t^2(x,a)}{(q^{P_t, \pi_t}(x,a) + \gamma)^2} \mathbb{E}_t \left[ \mathbb{1}_t(x,a) \right] \right)$$

$$= \frac{\ell_t^2(x,a) q^{P_t, \pi_t}(x,a)}{(q^{P_t, \pi_t}(x,a) + \gamma)^2}$$

$$\leqslant \frac{\ell_t^2(x,a)}{(q^{P_t, \pi_t}(x,a) + \gamma)}$$

$\square$

## F.2 PROPERTIES OF AVERAGE SMOOTHING

**Lemma F.3** (Bound on Smoothed Transitions). *Let $\overline{P}_t(\cdot|x,a) = \frac{1}{t} \sum_{t'=1}^{t} P_{t'}(\cdot|x,a)$, then $\left\| \overline{P}_t(\cdot|x,a) - \overline{P}_{t-1}(\cdot|x,a) \right\|_1 \leqslant \frac{2}{t}, \ \forall t \in [\![T]\!], \forall x,a \in \mathcal{X} \times \mathcal{A}$.*

*Proof.*

$$\left\| \overline{P}_t(\cdot|x,a) - \overline{P}_{t-1}(\cdot|x,a) \right\|_1 = \left\| \frac{1}{t} \sum_{t'=1}^{t} P_{t'}(\cdot|x,a) - \frac{1}{t-1} \sum_{t'=1}^{t-1} P_{t'}(\cdot|x,a) \right\|_1$$

$$\leqslant \left\| \frac{t-1-t}{t(t-1)} \sum_{t'=1}^{t-1} P_{t'}(\cdot|x,a) \right\|_1 + \left\| \frac{1}{t} P_t(\cdot|x,a) \right\|_1$$

$$\leqslant \frac{1}{t(t-1)} \sum_{t'=1}^{t-1} 1 + \frac{1}{t} = \frac{2}{t}$$

Where the first inequality comes from triangle inequality and the second from the fact we are dealing with elements of simplexes. $\square$

**Lemma F.4** (Bound on $\epsilon_{t,k}$). *For $\epsilon_{t,k} = \frac{1}{2} \| q_t^{\circ,k} - q_{t-1}^{\circ,k} \|_1$ and $\overline{P}_t = \frac{1}{t} \sum_{t'=1}^{t} P_t$ we have that,*

$$\epsilon_{t,k} \leqslant \epsilon_t = \frac{L}{t}$$

*Proof.*

$$\epsilon_{t,k} = \frac{1}{2} \left\| q_t^{\circ,k} - q_{t-1}^{\circ,k} \right\|_1$$

$$= \frac{1}{2} \sum_{x \in \mathcal{X}_k} \sum_{a \in \mathcal{A}} \sum_{x' \in \mathcal{X}_{k+1}} \left\| q_t^{\circ,k}(x,a,x') - q_{t-1}^{\circ,k}(x,a,x') \right\|_1$$

$$= \frac{1}{2} \sum_{x \in \mathcal{X}_k} \sum_{a \in \mathcal{A}} \sum_{x' \in \mathcal{X}_{k+1}} \left| q_t^{\circ,k}(x,a) \overline{P}_t(x'|x,a) - q_{t-1}^{\circ,k}(x,a) \overline{P}_{t-1}(x'|x,a) \right|$$

$$= \frac{1}{2} \sum_{x \in \mathcal{X}_k} \sum_{a \in \mathcal{A}} \sum_{x' \in \mathcal{X}_{k+1}} \left| q_t^{\circ,k}(x,a)\overline{P}_t(x'|x,a) - q_{t-1}^{\circ,k}(x,a)\overline{P}_{t-1}(x'|x,a) \pm q_{t-1}^{\circ,k}(x,a)\overline{P}_t(x'|x,a) \right|$$

$$\leqslant \frac{1}{2} \sum_{x \in \mathcal{X}_k} \sum_{a \in \mathcal{A}} \sum_{x' \in \mathcal{X}_{k+1}} \left| q_t^{\circ,k}(x,a)\overline{P}_t(x'|x,a) - q_{t-1}^{\circ,k}(x,a)\overline{P}_t(x'|x,a) \right|$$

$$+ \sum_{x \in \mathcal{X}_k} \sum_{a \in \mathcal{A}} \sum_{x' \in \mathcal{X}_{k+1}} \left| q_{t-1}^{\circ,k}(x,a)\overline{P}_t(x'|x,a) - q_{t-1}^{\circ,k}(x,a)\overline{P}_{t-1}(x'|x,a) \right|$$

$$\leqslant \frac{1}{2} \sum_{x \in \mathcal{X}_k} \sum_{a \in \mathcal{A}} \left| q_t^{\circ,k}(x,a) - q_{t-1}^{\circ,k}(x,a) \right| \sum_{x' \in \mathcal{X}_{k+1}} \overline{P}_t(x'|x,a)$$

$$+ \sum_{x \in \mathcal{X}_k} \sum_{a \in \mathcal{A}} q_{t-1}^{\circ,k}(x,a) \sum_{x' \in \mathcal{X}_{k+1}} \left| \overline{P}_t(x'|x,a) - \overline{P}_{t-1}(x'|x,a) \right|$$

$$\overset{a}{\leqslant} \frac{1}{2} \sum_{x \in \mathcal{X}_k} \sum_{a \in \mathcal{A}} \left| q_t^{\circ,k}(x,a) - q_{t-1}^{\circ,k}(x,a) \right| \sum_{x' \in \mathcal{X}_{k+1}} \overline{P}_t(x'|x,a) + \frac{1}{t}$$

$$\overset{b}{\leqslant} \frac{1}{2} \sum_{x \in \mathcal{X}_k} \left| q_t^{\circ,k}(x) - q_{t-1}^{\circ,k}(x) \right| + \frac{1}{t}$$

$$\overset{c}{\leqslant} \frac{k}{t} + \frac{1}{t} = \frac{k+1}{t} \leqslant \frac{L}{t}$$

Where "a" comes from Lemma F.3 and "b" from Lemma F.8 and finally "c" from Lemma F.9. □

### F.3 KNOWN LEMMAS AND COROLLARIES

**Lemma F.5** (Lemma D.3.3, Jin et al. 2023). *Denote the set of tuples $\mathcal{X}_k \times \mathcal{A} \times \mathcal{X}_{k+1}$ by $W_k$. For any transition functions $\overline{P}_t, P_t$, and any policy $\pi$,*

$$q^{\overline{P}_t,\pi}(x) - q^{P_t,\pi}(x) = \sum_{k=0}^{k(x)-1} \sum_{(u,v,w) \in W_k} q^{\overline{P}_t,\pi}(u,v)(\overline{P}_t(w|u,v) - P_t(w|u,v))q^{P_t,\pi}(x|w)$$

$$= \sum_{k=0}^{k(x)-1} \sum_{(u,v,w) \in W_k} q^{P_t,\pi}(u,v)(\overline{P}_t(w|u,v) - P_t(w|u,v))q^{\overline{P}_t,\pi}(x|w)$$

*where $q^{P',\pi}(x|w)$ is the probability of visiting $x$ under $\pi$ executed in $P$.*

According to Lemma F.5 we can estimate the occupancy measure difference caused by the error on the transition function at episode $t$ with the following corollary.

**Corollary F.6** (Corollary D.3.6, Jin et al. 2023). *For any episode $t$ and any policy $\pi$ we have:*

$$\left| q^{\overline{P}_t,\pi}(x) - q^{P_t,\pi}(x) \right| \leqslant C^{\overline{P}_t}, \ \forall x \neq x_L, \ and \ \sum_{x \neq x_L} \left| q^{\overline{P}_t,\pi}(x) - q^{P_t,\pi}(x) \right| \leqslant L C^{\overline{P}_t}$$

**Corollary F.7** (Corollary D.3.7, Jin et al. 2023). *For any policy sequence $\{\pi_t\}_{t=1}^T$, and loss functions $\{\ell_t\}_{t=1}^T$ such that $\ell_t : \mathcal{X} \times \mathcal{A} \to [0,1]$ for any $t \in \{1, \ldots, T\}$ it holds that*

$$\sum_{t=1}^T \langle q^{\overline{P}_t,\pi_t} - q^{P_t,\pi_t}; \ell_t \rangle \leqslant L C^{\overline{P}}$$

**Lemma F.8** (Lemma D.2, Rosenberg & Mansour 2019). *Let $\pi$ be a policy and let $\overline{P}_t, \overline{P}_{t-1}$ be transition functions such that $\|\overline{P}_t(\cdot|x,a) - \overline{P}_{t-1}(\cdot|x,a)\| \leqslant \nu, \ \forall x, a \in \mathcal{X} \times \mathcal{A}$ then the following equations hold,*

$$\sum_{x \in \mathcal{X}} \left| q^{\overline{P}_t,\pi}(x) - q^{\overline{P}_{t-1},\pi}(x) \right| = \sum_{x \in \mathcal{X}} \sum_{a \in \mathcal{A}} \left| q^{\overline{P}_t,\pi}(x,a) - q^{\overline{P}_{t-1},\pi}(x,a) \right|$$

$$\sum_{x \in \mathcal{X}} \sum_{a \in \mathcal{A}} \left| q^{\overline{P}_t,\pi}(x,a) - q^{\overline{P}_{t-1},\pi}(x,a) \right| \leqslant \sum_{x \in \mathcal{X}} \sum_{a \in \mathcal{A}} \sum_{x' \in \mathcal{X}_{k(x)+1}} \left| q^{\overline{P}_t,\pi}(x,a,x') - q^{\overline{P}_{t-1},\pi}(x,a,x') \right|$$

$$\sum_{x \in \mathcal{X}} \sum_{a \in \mathcal{A}} \sum_{x' \in \mathcal{X}_{k(x)+1}} \left| q^{\overline{P}_t, \pi}(x, a, x') - q^{\overline{P}_{t-1}, \pi}(x, a, x') \right| = \sum_{x \in \mathcal{X}} \sum_{a \in \mathcal{A}} \left| q^{\overline{P}_t, \pi}(x, a) - q^{\overline{P}_{t-1}, \pi}(x, a) \right| + L\nu$$

**Lemma F.9** (Lemma E.2, Rosenberg & Mansour 2019). *Let $\pi$ be a policy and let $\overline{P}_t, \overline{P}_{t-1}$ be transition functions such that $\left\| \overline{P}_t(\cdot|x, a) - \overline{P}_{t-1}(\cdot|x, a) \right\| \leqslant \nu, \ \forall x, a \in \mathcal{X} \times \mathcal{A}$. Then $\forall k \in [\![0, L-1]\!]$*

$$\sum_{x_k \in \mathcal{X}_k} \left| q^{\overline{P}_t, \pi}(x_k) - q^{\overline{P}_{t-1}, \pi}(x_k) \right| \leqslant k\nu$$

**Corollary F.10** (Corollary E.2, Rosenberg & Mansour 2019). *Let $\pi$ be a policy and let $\overline{P}_t, \overline{P}_{t-1}$ be transition functions such that $\left\| \overline{P}_t(\cdot|x, a) - \overline{P}_{t-1}(\cdot|x, a) \right\|_1 \leqslant \nu, \ \forall x, a \in \mathcal{X} \times \mathcal{A}$. Then $\forall k \in [\![0, L-1]\!]$*

$$\left\| q^{\overline{P}_t, \pi} - q^{\overline{P}_{t-1}, \pi} \right\|_1 \leqslant L^2 \nu + L\nu = \mathcal{O}(L^2 \nu)$$

