# OpenReview forum: "No-regret Learning with Revealed Transitions in Adversarial Markov Decision Processes"
_ICLR.cc/2025/Conference — ICLR 2025 Conference Withdrawn Submission_

### Official Review · Reviewer_nix3 · 2024-10-28

**Soundness:** 3
**Presentation:** 2
**Contribution:** 2
**Rating:** 5
**Confidence:** 4

**Summary:**

Summary: This paper considers the setting where the rewards and transitions are adversarial, and the transition kernel will be revealed to the agent at the end of each round. In this new setting, authors introduce the concepts of smoothed MDPs and smoothed regret. In particular, the smoothed MDP is defined as a transition kernel aggregating with previously experienced ones. The authors propose an OMD-based algorithm, called SOMD, to achieve $O(L^{3/2}\sqrt{T})$ smoothed regret bound for full-information feedback. Further, SOMD is extended to handle the bandit feedback.

**Strengths:**

1. This paper introduces interesting concepts of smoothed MDP and smoothed regret. The new transition model is particularly interesting since it subsumes two models studied in prior works as special cases. I appreciate the authors’ efforts to highlight their connections.

2. The authors develop new algorithms for the smoothed MDP with full information and bandit feedback. Theoretical guarantees are provided w.r.t. smoothed regret and other regret by translating their smooth regret bounds.

3. The description of algorithms is clear and easy to follow. The related work is also well categorized.

**Weaknesses:**

1. The transition is revealed at every round is a pretty strong assumption. This assumption significantly simplifies AMDPs problem since the main challenge is to deal with an unknown and adversarial transition model. In this case, the algorithm does not handle the varying transitions but instead exploits the revealed transitions.

2. The paper does not provide a clear motivation for considering revealed transitions. For example, in the abstract, there seems to be no direct link between the first two sentences. While the problem of adversarial transitions and rewards is central to learning AMDPs, it is not immediately clear why this motivates the study of a weaker model with revealed transitions.

3. The algorithmic contribution seems somewhat limited, as similar ideas have been explored in previous work, particularly in the context of [1].

[1] Mirror Descent Meets Fixed Share (and feels no regret)

**Questions:**

1. Are there any motivating examples to consider the revealed transition model?

2. It looks like the main difficulty to acquiring any meaningful bound when transitions are not revealed is the function $f_t$ is not specified. It could be interesting to try to get some bound by letting the function class (of $f_t$) be as large as possible, and the transitions are unknown.

---

### Official Review · Reviewer_uJGC · 2024-10-30

**Soundness:** 3
**Presentation:** 3
**Contribution:** 1
**Rating:** 3
**Confidence:** 4

**Summary:**

This paper considers the MDPs where rewards and transitions are both adversarial. While it is statistically or computationally impossible to achieve sub-linear regret in general, this paper considers the case where we are given full-information feedback on the adversarial transitions and the transitions allows a low $\ell_\infty$ variation w.r.t. some reference transition model.

By pretending that the transition model is the average of all historical transitions $\frac 1t \sum_{\tau \le t} P_\tau$ and perform an EXP3 on the occupancy measure space induced by this "smoothened" transition model, the authors yield algorithms ensuring $\tilde{\mathcal O}(\sqrt T + C^P)$ regret in both full-information or bandit feedback models (for rewards).

**Strengths:**

1. The idea of utilizing feedback on transitions to tackle the time-varying transition model is intuitive and well-explained.
2. The results enjoy a nice dependency on both $T$ and $C^P$.

**Weaknesses:**

While, as I said in the Strengths, the idea of utilizing feedback on transitions to tackle the time-varying transition model is intuitive and well-explained, it also makes me unsurprised about the algorithms and the results. Given that $\sum_t \lVert P_t - P\rVert \le C$, it is straightforward that the average transitions would converge to the reference transition and thus allowing good approximation effect.

Besides the smoothened OMD & regret part, the other techniques (like occupancy measures or implicit exploration) are pretty standard for adversarial MDPs. In terms of the smoothened part -- because of the well-established research on occupancy measures -- it is also not hard to translate the smoothened regret to the standard regret. I also do not see how this idea can be generalized to more challenging problems where the transition feedback are imperfect or the variation metric are different, or to other RL theory problems.

Therefore, at least based on the current main text, this paper lacks technical contributions to be of general interest. The problem setup, albeit indeed new and different from previous ones, is also insufficient to make this paper stand out. This is the reason why I vote for a reject. If I am missing something, please let me know and I will be happy to re-evaluate this paper.

**Questions:**

Please refer to the Weakness part.

---

### Official Review · Reviewer_mRDn · 2024-11-02

**Soundness:** 3
**Presentation:** 3
**Contribution:** 2
**Rating:** 5
**Confidence:** 3

**Summary:**

This paper studies learning MDPs with adversarially chosen losses and transitions. The loss could be either full-information or bandit feedback, but the transition is revealed at the end of each round. The authors propose smoothed transition error $\bar{C}^P_f$ as a more general measure for the changing transitions and derive sublinear regret bound with an additive term of $\bar{C}^P_f$. A new regret decomposition and a new technique smoothed OMD is proposed to achieve this.

**Strengths:**

1. This paper studies an important online learning problem where both reward and transition can be adversarially chosen.
2. This paper proposes a new measure of changing transitions and shows that it is more general than previous measures.
3. This paper proposes a new  technique smoothed OMD to help the analysis.

**Weaknesses:**

1. Compared with [Jin et al.,2023], this paper requires the transition to be revealed at the end of every round, which is a strong assumption.
2. While the paper introduces an additional assumption and presents a more general framework, it appears to offer limited new learnability results compared to [Jin et al., 2023]. The current interesting regret results are all based on the average smoothing function, which leads to almost the same regret compared with [Jin et al., 2023] given Example 3.2. Although Theorem 4.1 provides a general regret bound, it is unclear whether this general theorem can lead to interesting sublinear regret with refined transition error measure compared with [Jin et al., 2023].

**Questions:**

1. If we choose other smoothing functions beyond the average one, is it possible to get a $\sqrt{T}$ regret with a better additive transition error measure compared with [Jin et al., 2023]? For example, is it possible to achieve this with the smoothing function in Example 3.1?

2. Could the authors provide more technical insights about the reason for choosing smoothed regularization at Line 4 of Algorithm 1?

---

### Official Review · Reviewer_xYMJ · 2024-11-03

**Soundness:** 2
**Presentation:** 2
**Contribution:** 2
**Rating:** 3
**Confidence:** 3

**Summary:**

This paper studies the problem of learning an adversarial MDP with revealed transitions. The authors propose a notion of smoothed MDP based on a sequence of generic function $\{f_t\}_{\tgeq 1}$, and devise Smoothed Online Mirror Descent (SOMD) to reach $\tilde{O}(\sqrt{T}+ \bar{C}^{P}_{f})$ regret, where $\bar{C}^{P}_{f}$ describe the variation of the transition model measured by $f$.

**Strengths:**

The SOMD algorithm could cooperate with different smooth functions, which is more flexible than the classical OMD algorithm.

**Weaknesses:**

1.  The overall contribution is incremental in the aspect of RL theory,  given that (Jin et. al., 2023) had established a similar regret bound without transition information.


2. It is not clear how to choose the smooth function $f_t$ efficiently to minimize the regret even if assuming the transition is revealed by the end of each episode.

**Questions:**

I did not go through the details but it seems weird if $f_{t}$ is a function of $(P_1,P_2,....,P_t)$. If so, we can let $C_f^P=0$  by choosing $f_t(P_1,P_2,...,P_t)=P_t$. So could I guess $f_t$ might be a function of $(P_1,P_2,...,P_{t-1})$?

---

### Note · Authors · 2024-11-20

**Comment:**

We thanks the Reviewers for the time spent for reviewing our paper. We decided to withdraw our submission and to strengthen our work.

Thank you again,

the Authors.

**Withdrawal Confirmation:**

I have read and agree with the venue's withdrawal policy on behalf of myself and my co-authors.